# TRUSTED MULTI-VIEW CLASSIFICATION VIA EVOLUTIONARY MULTI-VIEW FUSION

**Xinyan Liang**[1,*] **Pinhan Fu**[1*], **Yuhua Qian**[1,†] **Qian Guo**[2], **Guoqing Liu**[1]

[1] Institute of Big Data Science and Industry, Key Laboratory of Evolutionary Science
  Intelligence of Shanxi Province, Shanxi University
[2] Shanxi Key Laboratory of Big Data Analysis and Parallel Computing,
  School of Computer Science and Technology, Taiyuan University of Science and Technology
`{liangxinyan48,fupinhan168,czguoqian,guoqingl1001}@163.com`
`jinchengqyh@126.com`

## ABSTRACT

Multi-view classification based on the Dempster-Shafer theory is widely recognized for its reliability in safety-critical domains with multi-view data. However, the adoption of a late fusion strategy constrains information interaction among views, thereby leading to suboptimal utilization of multi-view data. A recent advancement addressing this limitation involves generating a pseudo view by concatenating individual views. Yet, the efficacy of this pseudo view may diminish when incorporating underperforming views like noisy views. Additionally, the integration of a pseudo view exacerbates the issue of imbalanced multi-view learning, as it contains a disproportionate amount of information compared to individual views. To address these issues, we propose the enhancing Trusted multi-view classification via Evolutionary multi-view Fusion (TEF) approach. TEF employs an evolutionary multi-view architecture search method to create a high-quality fusion architecture serving as the pseudo view, facilitating adaptive view and fusion operator selection. Furthermore, TEF enhances each view within the fusion architecture by concatenating the fusion architecture's decision output with its respective view. Our experimental results demonstrate the effectiveness of this straightforward yet powerful strategy in mitigating imbalanced multi-view learning issues, particularly on complex many-view datasets exceeding three views. Extensive evaluations across 13 multi-view datasets validate the superior performance of our proposed method compared to other trusted multi-view learning approaches. The code is available at `https://github.com/fupinhan123/TEF`.

## 1 INTRODUCTION

Multi-view classification (MVC) endeavors to construct classifiers utilizing multi-view data, wherein each sample is characterized by multiple groups of feature sets (Liang et al., 2022; Wen et al., 2024; Jiang et al., 2024; Zhang et al., 2024a; Lyu et al., 2024b; Jiang et al., 2021). Notably, a plethora of methodologies has been proposed in this domain. Among these, trusted multi-view classification (TMVC) methods (Xu et al., 2024; Liu et al., 2023) deviate from conventional approaches by leveraging distinct views at the level of evidence rather than focusing solely on features. They have garnered significant application across various safety-critical medical diagnostic fields, including retinal anomaly identification (Wang et al., 2023), eye disease screening (Zou et al., 2023), and medical image classification (Xu et al., 2022).

However, the adoption of a late fusion strategy restricts the interaction of information among views, leading to suboptimal utilization of multi-view data. To address this limitation,Han et al. (2023) proposed an enhanced trusted multi-view classification (ETMC). ETMC augmented their pioneering work on Trusted Multi-view Classification (TMC) (Han et al., 2021) by introducing a pseudo view obtained through the concatenation of individual views, thereby enriching it with complementary

---

*Equal contribution
†Corresponding author

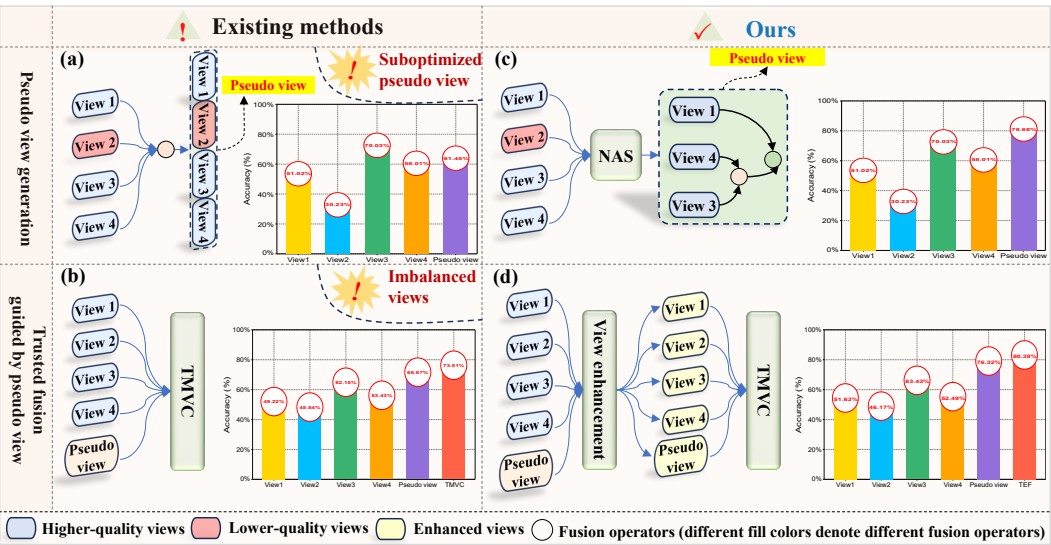

Figure 1: Two issues of existing trusted multi-view learning and our solutions.

information. Yet, real-world multi-view data often suffer from noise or uncertainty due to complexities in data collection and transmission (Zhou et al., 2023). Additionally, each view may have its own feature or distribution, in some cases, the data in each separate view may not be compatible with the other. Obviously, if a fusion approach is unable to cope appropriately with multiple views, the obtained pseudo view may be suboptimized when incorporating underperforming views. This issue is illustrated in Fig. 1(a) where the classification accuracy of the pseudo view is lower than the view 3. The similar results on experimental multi-view datasets can be observed by comparing the accuracy of pseudo views from Table 1 with one of best view (BV) from Table 3. Regrettably, the pseudo view generation strategy of ETMC is susceptible to negative influences from noisy or corrupted views in certain real-world scenarios.

Moreover, as shown in Fig. 1(b), the pseudo view inherently contains a disproportionate amount of information compared to individual views. The phenomenon is referred to as imbalanced views. Recent study (Peng et al., 2022) has shown that the multi-view models learn on imbalanced views may be suboptimized evens fail to outperform its uni-view counterpart due to view with better performance contributes to lower whole loss then dominates the optimization progress via propagating limited gradient over the other view. To our knowledge, the exiting pseudo-view-guided trusted multi-view learning have not yet paid attention to this issue, limiting their performance. In addition, the predominant approach to tackling imbalanced multi-view learning involves the manipulation of gradients and they are often a challenging technique to use (Wei et al., 2024; Wang et al., 2020).

To address these issues, we propose the enhancing Trusted multi-view classification via Evolutionary multi-view Fusion (TEF) approach. Specifically, as shown in Fig.1(c), TEF introduces an evolutionary multi-view architecture search method to generate a high-quality fusion architecture serving as the pseudo view, enabling adaptive selection of views and fusion operators. To mitigate the imbalanced multi-view learning problem, each view within the fusion architecture is enhanced by concatenating the decision output of the fusion architecture with its respective view shown in Fig.1(d). With the two simple but effective techniques, the potential of trusted multi-view classification is significantly unleashed, particularly on complex many-view datasets featuring more than three views compared to its counterparts. Notably, compared to gradient-based NAS that requires a predefined search space and substantial memory, and reinforcement learning-based NAS that relies on extensive computational resources, evolutionary NAS offers advantages such as global search capability, flexibility, and parallelization (Liang et al., 2021; Fu et al., 2025). This makes it particularly suitable for handling complex multi-view tasks and large search spaces. The issue of time consumption for TEF will be discussed and addressed in the experimental section 4.2. The contributions of this paper are:

- The pseudo view generation is formulated as a population-based multi-view neural architecture problem where the higher-quality views can be automatically selected and fused with selected some fusion operators from a candidate fusion operator set.

- From the perspective of view enhancement, we propose a simple yet effective strategy to deal with the issue of imbalanced multi-view learning that is raised by a disproportionate amount of information between pseudo view and individual views.

- Extensive evaluations across 13 multi-view datasets validate the superior performance of our proposed method compared to other trusted multi-view learning approaches.

## 2 RELATED WORK

**Multi-view classification (MVC):** MVC methods can be divided into early fusion and late fusion Guo et al. (2024); Lyu et al. (2024a); Zhang et al. (2024b). Early fusion combines data from different views at the initial stage to form a unified representation before inputting it to the classifier. Techniques include element-wise addition, multiplication, and feature concatenation, along with advanced operators like MLB (Kim et al., 2017), MFB (Yu et al., 2018), TFN (Zadeh et al., 2017), LMF (Liu et al., 2018), and PTP. This approach fully utilizes information from all views but may suffer from dimensionality issues and redundancy. Late fusion processes each view independently and merges the outputs of multiple classifiers using methods like majority voting and weighted averaging. It offers strong model flexibility and reduces view interference but has higher decision fusion complexity. Early fusion focuses on feature-level integration, while late fusion emphasizes decision-level integration, each with its own strengths and weaknesses. As shown in Fig. 2, our TEF possesses their strengths by achieving early fusion and late fusion at the stage 1 and the stage 2, respectively.

**Trusted multi-view classification (TMVC) :** TMVC, a late-fusion method, stands out for its ability to integrate diverse views based on their respective trustworthiness, quantified through the Dempster-Shafer theory (Xu et al., 2024; Liu et al., 2023). A typical example is TMC (Han et al., 2021), which involves Dempster's combination rule that assigns smaller weights to highly uncertain views. Following this approach, various opinion aggregation methods have been proposed (Liu et al., 2022; Han et al., 2023). Recent studies have highlighted that a significant feature of previous methods is that the uncertainty mass decreases after integrating another opinion into the original one. Recently, RCML introduced the idea that uncertainty should increase when incorporating unreliable or conflicting opinions. To address this, RCML (Xu et al., 2024) proposed a conflictive opinion aggregation strategy and theoretically proved that uncertainty increases for conflicting instances. TMVC methodologies differ from traditional approaches by leveraging distinct views at the evidence level rather than focusing solely on features. The utilization of the variational Dirichlet distribution is pivotal in modeling the distribution of class probabilities. This distribution, parameterized with evidence gleaned from diverse views, facilitates the integration of evidence through the interpretable framework of the Dempster-Shafer theory. This strategic shift yields a more stable and reasonable estimation of uncertainty, thereby enhancing the reliability and robustness of classification outcomes. Although they have shown great potential for widespread adoption in safety-critical domains, they sometimes fail to outperform non-trusted multi-view classification method even uni-view counterpart due to the limited interaction among views and imbalanced issue among views.

## 3 METHOD

The enhancing trusted multi-view classification via evolutionary multi-view fusion (TEF) approach consists of two stages: (1) obtaining a high-quality fusion architecture as pseudo view with neural architecture search, and (2) balanced and trusted fusion driven by enhanced pseudo view. Its overview is illustrated in Fig. 2.

### 3.1 PSEUDO VIEW GENERATION

The research conducted by (Han et al., 2023) underscores the significance of pseudo views in augmenting the performance of TMC method. Particularly, early interaction among views is identified as a critical factor in this enhancement. To refine the quality of pseudo views, we advocate for

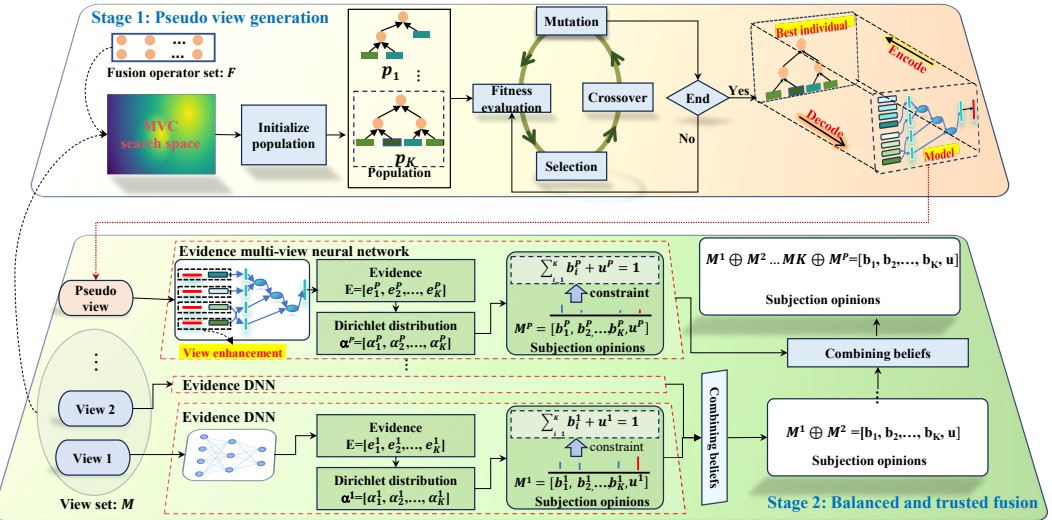

Figure 2: The overview of TEF

an automated approach. Our proposition involves framing the pseudo view generation process as a multi-view neural architecture search task within the evolutionary computation paradigm. This methodological framework holds promise for eliciting superior pseudo views, thus offering a pathway to further elevate the efficacy of the trusted multi-view learning.

**Multi-view classification model encoding and individual decoding:** Specifically, let $V$ denote the set of views and $F$ represent the set of fusion operators (see A.2). These sets collectively define a multi-view neural architecture space, denoted as $H$. Within our framework, each multi-view classification model $h$ from $H$ corresponds to an individual in the evolutionary computation framework. It can be encoded multiple forms like tuple or tree. The study on individual encode is out of scope of our work. Here, we take a tuple form for example to illustrate the process of encoding and decoding. Comparable results between different forms can be found in Section A.5 in the appendix. Notably, $h$ can be encoded as tuples $p = [v, f]$, where $v = [v_1, v_2, \cdots, v_l]$ represents $l$ used views, $f = [f_1, f_2, \cdots, f_{l-1}]$ denotes the utilized fusion schemes, and $v_i$ and $f_i$ are an element from $V$ and $F$, respectively.

Each individual chromosome vector $p$ can be decoded to a multi-view classification model $h$ that takes $v = [v_1, v_2, \cdots, v_l]$ as its input. Specifically, each $v_i$ is first transferred to $u_i$ by a fully connected layer and then a Relu function. Next fuse $z_1, z_2, \cdots, z_l$ in a simple yet effective behavior:

$$c_i = \begin{cases} f_i(z_1, z_2), & if \ i = 1, \\ f_i(c_{i-1}, z_{i+1}), & if \ i \in [2, 3, \cdots, l-1]. \end{cases} \quad (1)$$

To alleviate the issue of wide fluctuations in the fused vector values due to different fusion operator usage in the process of fusion, we normalize $c$ with the following process (Liang et al., 2021)

$$c \leftarrow \text{sign}(c) \mid \sqrt{|c|} \mid, \quad (2)$$

$$c \leftarrow \frac{c}{\| c \|}, \quad (3)$$

where sign denotes the sign function. Eq. (2) represents Power Law Normalization, while Eq. (3) denotes L2 Normalization. The effectiveness of this strategy has been validated by many works (Liang et al., 2021). Lastly, transfer $c$ to a probability vector $\hat{y}$ by a fully connected layer and a softmax function. The optimization objective adopts cross entropy loss and is defined as:

$$L_{ce} = \frac{1}{N} \sum_{i=1}^{N} \sum_{k=1}^{K} y_{ik} \log \hat{y_{ik}}, \quad (4)$$

where $N$ is the number of the examples, $K$ is the number of the categories, $y_{ik}$ is 1 if the $i^{th}$ example belongs to $k^{th}$ category, 0 otherwise, and $\hat{y_{ik}}$ is the prediction probability that the $i^{th}$ example belong to $k^{th}$ category, which is the output of the softmax layer.

**Optimization:** Our pseudo view, conceived as a multi-view neural architecture rather than a singular fusion vector, serves as a pivotal element in our approach. The optimal pseudo view can be autonomously searched via an evolutionary computation method. This search method, detailed in Alg. 1, encompasses population initialization, fitness evaluation, offspring generation, and selection, facilitating the systematic exploration and refinement of multi-view neural architectures.

In the initial stage of our evolutionary algorithm, population initialization involves randomly generating $N$ individual chromosome vectors, each potentially having a distinct value for $k$. Subsequently, the fitness of each chromosome vector $p$ within the population is assessed by decoding it into a multi-view classification model. This model is trained on a multi-view dataset and its classification accuracy is measured on a separate validation set, serving as the fitness metric for $p$. More details can be found in Section A.7 in the appendix.

## 3.2 BALANCED AND TRUSTED FUSION

This part aims to make a better trusted making-decision by jointly fusing the all views and the generated pseudo view. It is nature idea to directly pass each view into an evidence-based classifier (Sensoy et al., 2018). Yet, the pseudo view inherently contains a disproportionate amount of information compared to individual views. Thus, its integration exacerbates concerns regarding imbalanced multi-view learning (Peng et al., 2022). We proposed a very effective solution by adding the output into the selected views. Specifically, our trusted fusion works as follows:

(1) View enhancement: Let $h$ be the optimal multi-view fusion model searched via Alg. 1, and its output is denoted as $h^{out}$. We enhance each view in $h$ by concatenating it with $h^{out}$, denoted as $[v_i, h^{out}]$, producing $v^{ep} = [[v_1, h^{out}], [v_2, h^{out}], \cdots, [v_k, h^{out}]]$.

(2) Evidence collection: Let $E = [e_1, e_2, \cdots, e_K]$ be the evidences where each element $e_k$ is the $k$-th category evidence and its value is larger than 0. Then it can be obtained by replacing the softmax activation function with the Softplus activation function:

$$E = \text{Softplus}(F^{out}), \tag{5}$$

where $F^{out}$ is the last layer of the backbone. It is noted that the backbone of the enhanced pseudo view is $h$ but its takes $v^{ep}$ as input instead of $v$.

(3) Dirichlet distribution calculation: Let $\alpha = [\alpha_1, \alpha_2, \cdots, \alpha_K]$ be the concentration parameters of one Dirichlet distribution where each element $\alpha_k$ is the $k$-th catagory Dirichlet distribution parameter and can be obtained via

$$\alpha_k = e_k + 1. \tag{6}$$

$\mu = [\mu_1, \mu_2, \cdots, \mu_K]$ be the mean of the corresponding Dirichlet distribution, its each element $\mu_k^m$ is computed as

$$\mu_k = \frac{\alpha_k}{S}, \tag{7}$$

where $S = \sum_{k=1}^{K} a_k$ is the Dirichlet intensities. Then the probability density function of the Dirichlet distribution is given by:

$$Dir(\mu|\alpha) = \frac{1}{B(\alpha)} \prod_{i=1}^{K} \mu_i^{\alpha-1}, \tag{8}$$

where $B(\alpha)$ is the $K$-dimensional multinomial beta function.

(4) Subjective opinions calculation. Let $b$ be a set of belief masses and $u$ is its corresponding uncertainty score. Then the subjective opinion $M$ can be represented as

$$M = [b, u], \tag{9}$$

where $b = [b_1, b_2, \cdots, b_K]$ where each element $b_k$ is the belief mass. Let $S = \sum_{k=1}^{K} a_k$ be the Dirichlet intensities. Then $b_i$ and $u$ can be obtained by

$$b_k = \frac{e_k}{S} = \frac{a_k - 1}{S}, u = \frac{K}{S}. \tag{10}$$

It can be seen from Eq. 10 the probability assigned to category $k$ is proportional to the observed evidence for category $k$.

(5) Fusion: Let $M^1, M^2, \cdots, M^V$ and $M^{sp}$ be $(V + 1)$ opinions obtained from $V$ views and the pseudo view. Then the trusted fusion is conducted via

$$M = \oplus_{i=1}^{V} M^i \oplus M^{sp}, \tag{11}$$

where $\oplus$ denotes the Dempster's combination operator for $K$-Class classification. Various implementation modalities are available like (Xu et al., 2024; Liu et al., 2023; Han et al., 2023). To show the advantage of our framework, we adopt its one simple implementation form from (Han et al., 2023). Specifically, given two opinions $M^1 = [b_1^1, b_2^1, \cdots, b_K^1, u^1]$ and $M^2 = [b_1^2, b_2^2, \cdots, b_K^2, u^2]$, we do the following fusion calculation to generate the combination opinion $M = [b_1, b_2, \cdots, b_K, u]$:

$$b_k = \frac{1}{1-C}(b_k^1 b_k^2 + b_k^1 u^2 + b_k^2 u^1), u = \frac{1}{1-C}u^1 u^2, \tag{12}$$

where $C = \sum_{i \neq j} b_i^1 b_j^2$.

(6) Probability of each class induction: With the Eq. 10, we can get the final probability of each class and the overall uncertainty via

$$S = \frac{K}{u}, e_k = b_k \times S. \tag{13}$$

**Optimization:** The objective function is

$$L = \sum_{i=1}^{N} \sum_{m=1}^{M} L^m(x_n^m, y_n) + \sum_{i=1}^{N} L^{pseudo}(\{x_n^m\}_{m=1}^M, y_n) + \sum_{i=1}^{N} L^{fused}(\{x_n^m\}_{m=1}^M, y_n), \tag{14}$$

where $L^m(x^m, y)$, $L^{pseudo}(\{x_n^m\}_{m=1}^M, y_n)$ and $L^{fused}$ are the variational loss functions for the $m$-th view, the pseudo view and final decision, and their definition are as follows

$$\begin{cases} L^m = E_{q_\theta^m(\mu^m|x^m)}[\log p(y|\mu^m)] - \lambda_t D_{KL}[Dir(\mu^m|\hat{\alpha}^m)||Dir(\mu^m|[1, \cdots, 1])], \\ L^{pseudo} = E_{\mu^p \sim Dir(\mu^p|\alpha^p)}[\log p(y|\mu^p)] - \lambda_t D_{KL}[Dir(\mu^p|\hat{\alpha}^p)||Dir(\mu^p|[1, \cdots, 1])], \\ L^{fused} = E_{\mu \sim Dir(\mu|\alpha)}[\log p(y|\mu)] - \lambda_t D_{KL}[Dir(\mu|\hat{\alpha})||Dir(\mu|[1, \cdots, 1])], \end{cases} \tag{15}$$

where $\lambda_t$ balances the expected classification error and KL regularization,

$$E_{q_\theta^m(\mu^m|x^m)}[\log p(y|\mu^m)] = E_{\mu^p \sim Dir(\mu^p|\alpha^p)}[\log p(y|\mu^p)] = \sum_{k=1}^{K} y_k(\psi(\alpha_k^m) - \psi(S^m)) \tag{16}$$

$$D_{KL}[Dir(\mu^m|\hat{\alpha}^m)||Dir(\mu^m|[1, \cdots, 1])] = \log(\frac{\Gamma(\sum_{k=1}^K \hat{\alpha}_k)}{\Gamma(K)\prod_{k=1}^K \Gamma(\hat{\alpha}_k)}) + \sum_{k=1}^{K}(\hat{\alpha}_k - 1)[\psi(\hat{\alpha}_k) - \psi(\sum_{j=1}^K \hat{\alpha}_j)] \tag{17}$$

where $q_\theta^m(\mu^m|x^m)$ is a probabilistic encoder with parameters $q_\theta^m$, $\psi(.)$ and $\Gamma(.)$ are the digamma function and gamma function, respectively; $y_k$ is the $k$-th element of $y$ that is a onehot vector. $S^m = \sum_{k=1}^K \alpha_k^m$ is the Dirichlet distribution $Dir(\boldsymbol{\mu}^m|\hat{\alpha}^m)$, $\hat{\alpha}^m = y + (1 - y) \odot \alpha^m$ is the adjusted parameter of the Dirichlet distribution to prevent penalizing evidence of the groundtruth class to 0.

The optimization process for the proposed model is summarized in Alg. 2.

## 4 EXPERIMENTS

In this section, we aim to validate the effective of TEF from six aspects: (1) Comparison with trusted multi-view classification methods, (2) comparison with other state-of-the-art methods, (3) impact analysis of different fusion strategies on pseudo-view quality, (4) impact analysis of different pseudo-view generation methods on TEF, (5) impact analysis of different strategies to address view imbalance, and (6) computation time analysis. Moreover, we offer a more in-depth analysis for our TEF by conducting more ablation experiments. These results can be found in the appendix A.5.

### 4.1 SETUP

We briefly present the experimental setup here, including the experimental datasets, evaluation metrics, model selection, and comparison methods.

**Datasets.** In our experiments, we cover six datasets with multiple views, namely Animals with Attributes (AWA), NUS-WIDE-128 (NUS), Reuters, MVoxCeleb, and YoutubeFace. For the Reuters dataset, we generate two versions by adding Gaussian noise, named Reuters5 and Reuters3. AWA and NUS contain seven views each, while Reuters, MVoxCeleb, and YoutubeFace have five views. For more detailed descriptions of the datasets, please refer to the appendix A.3.

**Evaluation metrics.** In the experiment, to avoid the randomness caused by data partitioning and network initialization, we adopted a 5-fold cross-validation strategy within the overall framework, dividing each dataset into training and testing sets. Notably, during the evolutionary search for the pseudo-view architecture, the training set was further split into a training set and a validation set to prevent data leakage. We reported the average performance and standard deviation across five data partitions. To evaluate the performance of TEF, we used five commonly used evaluation metrics: accuracy ($AC$), precision ($PE$), recall ($RE$), $F1$ score, and Kappa ($K$). Higher values of these evaluation metrics indicate better performance. All metrics were measured on a single P100 GPU. For more detailed information about the experimental environment, please refer to Appendix A.4.

**Comparison methods.** We compare TEF with three categories of multi-view classification methods (MVCs): (1) Trusted MVCs: For example, TMC, which integrates multi-view information at an evidence level using the Dempster-Shafer theory in a learnable manner. TMOA integrates multi-view information at an evidence level using opinion aggregation. ETMC improves upon previous methods by adding a pseudo view containing complementary information from multiple views and simply concatenating the original features. RCML fuses view-specific opinions consisting of decision results and reliability using a conflictive opinion aggregation strategy and simple and effective average pooling fusion. (2) Other state-of-the-art MVCs: such as BV, SSV, MR, EmbraceNet, AWDR, RMA, and DC-NAS (Liang et al., 2024). (3) Various multi-view fusion strategies includeing five basic fusion operators: sum, mean, max pooling, product, and concatenation as well as five advanced fusion operators: MLB, MFB, TFN, LMF, and PTP. For a detailed description of these methods, please refer to the appendix A.1.

### 4.2 EXPERIMENTAL RESULTS

**Comparison with trusted multi-view classification methods.** According to the results in Table 1, TEF demonstrates significant advantages over various advanced trusted fusion methods across five metrics on six datasets. For instance, in the AWA, NUS, and VoxCeleb datasets, TEF achieves approximately 4.09%, 2.07%, and 3.71% accuracy improvements, respectively, compared to the second-ranked model. This is attributed to the effective implementation of feature-level information interaction achieved by the high-quality pseudo views obtained through adaptive selection. This approach addresses the issue of view imbalance caused by high-quality fusion views, thereby enhancing the model's learning capability. This has been validated in the ablation study.

**Comparison with other state-of-the-art (SOTA) methods.** We further compare the TEF with other SOTA methods including BV, SSV, MR, EmbraceNet, AWDR, RMAR, and DC-NAS (Liang et al., 2024). The conclusion can be made based on the results in Table 1: (1) Performance superiority: While these SOTA methods outperform previous trusted fusion methods in terms of raw performance metrics, they fall short in providing trustworthy decisions at the view fusion level, thus lacking in security and reliability. (2) Trustworthy decision: TEF, as a trusted fusion architecture, not only

Table 1: Accuracy comparison results with trusted MVCs and other SOTA MVCs (mean $\pm$ standard deviation), where the best performance is highlighted in bold. The other four metrics are provided Table 12 in the appendix.

| Methods | AWA | NUS | Reuters5 | Reuters3 | VoxCeleb | YoutubeFace |
|---|---|---|---|---|---|---|
| TMC (ICLR22) | 88.59±0.25 | 72.73±0.30 | 79.60±0.56 | 84.23±0.35 | 73.13±0.15 | 71.18±2.27 |
| TMOA (AAAI22) | 89.17±0.31 | 72.60±0.48 | 79.11±0.43 | 84.19±0.27 | 84.72±0.21 | 84.35±0.25 |
| ETMC (TPAMI23) | 88.24±0.17 | 73.05±0.67 | 79.80±0.41 | 84.25±0.42 | 88.70±0.15 | 79.63±1.89 |
| RCML (AAAI24) | 89.06±0.21 | 72.53±0.55 | 81.39±0.18 | 85.88±0.29 | 80.51±0.41 | 81.95±0.20 |
| BV | 88.65±0.43 | 68.69±0.59 | 80.61±0.25 | 83.98±0.14 | 63.25±0.14 | 82.01±0.18 |
| SSV | 82.37±1.26 | 63.70±0.64 | 79.51±0.41 | 84.71±0.22 | 85.10±0.23 | 84.43±0.31 |
| MR | 87.10±0.64 | 64.39±0.85 | 78.24±0.45 | 84.17±0.19 | 79.92±0.29 | 84.78±0.21 |
| EmbraceNet | 84.97±0.23 | 72.43±0.38 | 80.07±0.21 | 83.58±0.25 | 81.74±0.34 | 80.90±1.04 |
| AWDR(PR19) | 90.46±0.06 | 72.44±0.66 | 79.69±0.27 | 83.32±0.32 | 91.08±0.09 | 85.11±0.15 |
| RMAR(INS22) | 90.63±0.13 | 72.51±0.67 | 79.84±0.25 | 83.48±0.25 | 91.54±0.11 | 85.21±0.17 |
| DC-NAS (AAAI24) | 90.66±0.15 | 74.35±0.58 | 81.35±0.28 | 85.86±0.14 | 92.19±0.07 | 85.28±0.14 |
| **TEF (Ours)** | **93.26±1.25** | **75.12±0.57** | **82.26±0.23** | **86.49±0.10** | **92.41±0.12** | **86.02±0.41** |

achieves superior performance metrics but also ensures trustworthy decisions at the view fusion level. This dual advantage sets TEF apart from other advanced methods. For instance, on the AWA and NUS datasets, TEF achieves approximately 2.6% and 0.77% accuracy improvements compared to the second-best model. These improvements highlight the significant advantage of TEF in ensuring both high performance and security in decision making. (3) The trustworthiness is derived from $1 - u$, where $u$ is the uncertainty score in Eq. 9. Compared to TMC and ETMC, we have significant advantages in making trustworthy decisions as shown in Table 2. For example, we achieved improvements of 26.9% and 15.6% on the AWA and NUS datasets, respectively. This is because the quality of pseudo-view data after sufficient feature interactions is high, leading to more reliable decisions and strengthening the overall architecture.

Table 2: Trustworthiness comparison across different datasets for TMC, ETMC and TEF.

| Datasets | TMC | ETMC | TEF (Ours) |
|---|---|---|---|
| AWA | 0.461±0.008 | 0.494±0.013 | **0.763±0.046** |
| NUS | 0.684±0.029 | 0.793±0.044 | **0.949±0.010** |
| Reuters5 | 0.908±0.011 | 0.842±0.015 | **0.959±0.002** |
| Reuters3 | 0.923±0.010 | 0.879±0.021 | **0.986±0.002** |
| YoutubeFace | 0.621±0.031 | 0.901±0.010 | **0.963±0.004** |

**Impact analysis of different fusion strategies on pseudo-view quality.** To investigate the importance of adaptive view selection for pseudo-view quality (PV), we compare our PV with five basic fusion operators and five advanced fusion operators. Table 3 shows their respective performances. It can be observed that each operator has its own advantages, with no single operator performing excellently across all datasets. For instance, except for PV, the Max fusion operator performs best on the AWA dataset, but performs mediocrely on other datasets. Additionally, although advanced fusion operators involve complex fusion operations, they sometimes perform lower than basic fusion operators in terms of performance. The strategy of adaptive view fusion achieves the best performance. These findings highlight the importance of adaptive view selection and fusion operators.

**Impact analysis of different pseudo-view generation methods on TEF.** This section aims to validate the impact of different pseudo-view generation methods on trusted multi-view learning. Specifically, five basic fusion operators are used as the pseudo views generation method at first stage of the proposed TEF, respectively. From Table 4, it is evident that the pseudo views using the single fusion operator are significantly inferior to the high-quality pseudo views obtained through our adopted NAS strategy. For instance, on the YoutubeFace dataset, our approach outperforms the second-best performing method by 7.66%, and similar improvements of at least one percentage point are observed on the remaining datasets. This fully demonstrates the importance of introducing high-quality pseudo views and underscores the necessity of adaptively selecting views and fusion operators to compose high-quality pseudo views.

Table 3: Accuracy comparison results with five basic fusion operators and five advanced fusion operators, the best performance is highlighted in boldface. "PV" denotes our pseudo view generation at the first stage in TEF. The other four metrics are provided Table 13 in the appendix.

| Methods | AWA | NUS | Reuters5 | Reuters3 | VoxCeleb | YoutubeFace |
|---------|-----|-----|----------|----------|----------|-------------|
| **Basic fusion operators** | | | | | | |
| Add | 88.56±0.08 | 72.81±0.70 | 79.70±0.25 | 83.46±0.28 | 87.53±0.41 | 82.40±0.23 |
| Mul | 86.53±0.41 | 64.58±0.63 | 77.02±0.38 | 81.89±0.70 | 72.31±0.90 | 83.18±0.14 |
| Cat | 88.22±0.17 | 72.32±0.50 | 79.91±0.28 | 83.66±0.17 | 87.98±0.20 | 83.05±0.56 |
| Max | 88.82±0.32 | 71.36±0.47 | 80.02±0.20 | 84.01±0.28 | 81.57±0.41 | 81.49±0.29 |
| Avg | 88.83±0.32 | 73.00±0.51 | 79.69±0.30 | 83.58±0.28 | 87.27±0.33 | 82.23±0.17 |
| **Advanced fusion operators** | | | | | | |
| MLB | 87.69±0.21 | 70.60±0.29 | 80.16±0.15 | 83.80±0.28 | 87.11±0.67 | 85.20±0.28 |
| MFB | 88.87±0.34 | 71.34±0.40 | 79.28±0.21 | 83.25±0.18 | 85.23±0.20 | 82.85±0.17 |
| TFN | 81.02±0.53 | 63.66±1.22 | 79.95±0.30 | 83.73±0.31 | 57.53±0.92 | 81.33±0.19 |
| LMF | 88.66±0.43 | 71.74±0.70 | 80.02±0.20 | 84.01±0.28 | 81.57±0.41 | 81.49±0.29 |
| PTP | 87.78±0.28 | 71.83±0.50 | 80.10±0.10 | 84.06±0.20 | 88.61±0.36 | 85.18±0.30 |
| **PV (Ours)** | **91.43±0.20** | **74.44±0.69** | **80.97±0.34** | **85.40±0.30** | **92.25±0.20** | **85.60±0.16** |

Table 4: Accuracy comparison results (mean ± standard deviation) with pseudo-views formed by different fusion strategies introduced in a trustworthy fusion framework, the best performance is highlighted in boldface. Among them, $Cat^0$ is the direct concatenation of all original features, while $Cat^1$ unifies all views to the same dimension before concatenation.

| Methods | AWA | NUS | Reuters5 | Reuters3 | VoxCeleb | YoutubeFace |
|---------|-----|-----|----------|----------|----------|-------------|
| $Cat^0$ | 88.24±0.17 | 73.05±0.67 | 79.80±0.41 | 84.25±0.42 | 88.70±0.15 | 79.63±1.89 |
| Add | 90.29±0.06 | 73.05±0.62 | 78.99±0.65 | 84.42±0.68 | 90.65±0.08 | 78.36±1.54 |
| Mul | 89.13±0.24 | 71.85±0.50 | 77.70±0.58 | 83.00±0.48 | 77.83±0.29 | 64.09±1.10 |
| $Cat^1$ | 89.22±0.09 | 73.29±0.63 | 79.25±0.49 | 84.11±0.59 | 77.82±0.29 | 70.68±1.50 |
| Max | 90.54±0.12 | 72.29±0.55 | 78.53±0.63 | 84.03±0.52 | 84.02±0.08 | 74.80±2.91 |
| Avg | 90.63±0.18 | 72.12±0.84 | 79.86±0.72 | 84.93±0.57 | 89.49±0.31 | 76.40±0.99 |
| **TEF (Ours)** | **93.26±1.25** | **75.12 ± 0.57** | **82.26±0.23** | **86.49±0.10** | **92.41±0.12** | **86.02±0.41** |

**Impact analysis of different strategies to address view imbalance.** This section aims to validate the effectiveness of view enhancement strategy for addressing view imbalance by comparing TEF with its three degraded versions. (1) $TEF_0$: Stage 1 is removed from TEF, i.e., degrading into TMC (Han et al., 2021), without introducing pseudo-views. (2) $TEF_1$: View Enhancement is removed from the TEF, i.e., only a high-quality pseudo-view generation architecture is introduced. (3) $TEF_2$: Directly using the decisions output of the fusion architecture generated by the evolutionary algorithm

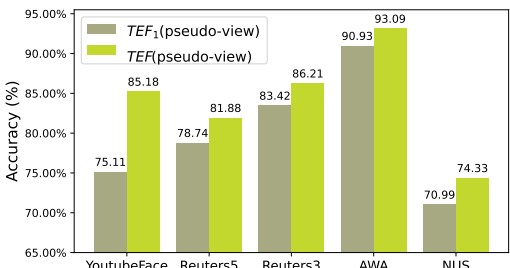
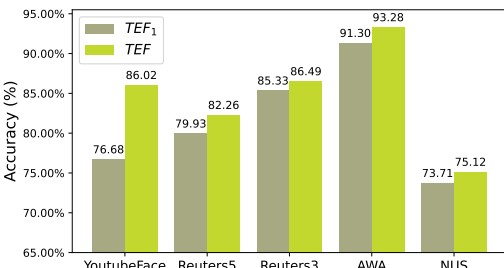

Figure 3: The accuracy comparison before and after solving the pseudo-view imbalance problem. The left chart shows the accuracy of pseudo-views in TEF, and the right chart shows the overall accuracy of TEF. The results of other metrics are shown in Fig. 8 in the appendix.

Table 5: Comparison results (mean $\pm$ std) on addressing the imbalance problem using feature enhancement versus directly using decision results, with the best performance highlighted in boldface.

| | Accuracy | | | | | |
|---|---|---|---|---|---|---|
| Methods | AWA | NUS | Reuters5 | Reuters3 | VoxCeleb | YoutubeFace |
| $TEF_0$ | 88.59$\pm$0.25 | 72.73$\pm$0.30 | 79.60$\pm$0.56 | 84.23$\pm$0.35 | 73.13$\pm$0.15 | 71.18$\pm$2.27 |
| $TEF_1$ | 91.60$\pm$0.20 | 73.71$\pm$0.48 | 79.93$\pm$0.54 | 85.33$\pm$0.41 | 91.44$\pm$0.14 | 76.68$\pm$1.44 |
| $TEF_2$ | 89.81$\pm$0.42 | 73.93$\pm$0.52 | 80.11$\pm$0.51 | 84.79$\pm$0.46 | 62.74$\pm$0.20 | 74.47$\pm$2.18 |
| **TEF** | **93.26$\pm$1.25** | **75.12 $\pm$ 0.57** | **82.26$\pm$0.23** | **86.49$\pm$0.10** | **92.41$\pm$0.12** | **86.02$\pm$0.41** |

Table 6: Time comparison of TEFs armed with different acceleration strategies on NUS dataset.

| Methods | FC | CS | Time |
|---|---|---|---|
| TEF1 | False | False | 13.21h |
| TEF2 | True | False | 10.35h |
| TEF3 | False | True | 4.43h |
| TEF4 | True | True | **2.46h** |

for trustworthy fusion, i.e., the decision output $h^{out}$ of Alg. 1 as the pseudo view. Observing Table 5 reveals the following findings: $TEF_1$ significantly outperforms $TEF_0$, highlighting the importance of introducing high-quality pseudo-views. The performance differences between $TEF_1$ and TEF indicate the presence of the view imbalance problem. For example, as shown in Fig. 3, introducing a high-quality pseudo-view generation architecture can significantly exacerbate imbalance issues, where the pseudo-views are not sufficiently trained. However, concatenated fusion decisions can significantly resolve this issue. For instance, in the YoutubeFace dataset, the performance improved from 75.11% without concatenation to 85.18% with concatenation. Similarly, TEF's performance increased from 76.68% to 86.02% with the introduction of concatenation. The results of $TEF_2$ indicate that directly introducing the decisions of the fusion architecture is ineffective. Despite its high individual accuracy, it cannot make a significant contribution to the overall trusted decision-making process, as it did not participate in the training of the entire trusted architecture.

**Computation time.** It is well known that evaluating the fitness of all $N$ chromosome vectors requires too much computing time at each generation. Fortunately, evolutionary multi-view learning has provided many accelerated solutions. In this part, we arm the TEF with two acceleration strategies. The first one is fitness caching (FC) (Liang et al., 2021) where previously evaluated vectors are recorded, avoiding redundant evaluations of individual chromosome vectors that have already been assessed. The second one is search guided by core structure (CS) (Fu et al., 2024a;b) where core structures are first found from a shrunk space, and then the optimal MMC model is found with the guidance of CSs from the expanded space. The results are shown in Table 6 where "True" and "False" denote that the corresponding acceleration strategies is used or not, respectively. One can see that the TEF4 that is the version of TEF armed with FC and CS is very high-efficiency. Obviously, new developments of the acceleration techniques in the evolutionary computation community can be readily integrated into the TEF, achieving a more higher-efficiency implementations of TEF if more effort is made.

## 5 CONCLUSION

In this paper, we have proposed an innovative approach to enhance the performance of the trusted multi-view classification. Our method has contributed to the advancement of trusted multi-view classification in two key dimensions. Firstly, we introduced a powerful mixture fusion framework that seamlessly integrates both early and late fusion techniques, with the early fusion process being automated for enhanced efficiency. Secondly, we addressed the challenge of imbalanced multi-view learning in large-scale datasets through a straightforward yet highly effective solution. By enriching the results of trusted multi-view learning through these novel strategies, our approach offers promising avenues for improved classification accuracy and scalability in complex real-world scenarios.

ACKNOWLEDGMENTS

This work was supported by National Natural Science Foundation of China (Nos. 62306171, T2495251, 62406218), the Science and Technology Major Project of Shanxi (No. 202201020101006), and Fundamental Research Program of Shanxi Province (No. 202203021222183).

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

# A APPENDIX

In the supplemental material:

- **A.1**: We describe the basic information of all comparative methods.
- **A.2**: We describe the five basic fusion operators used in evolutionary NAS.
- **A.3**: We give detailed descriptions of the datasets.
- **A.4**: We explain the implementation details.
- **A.5**: We present supplementary results.
- **A.6**: Comparison on benchmark datasets used by other TMVC methods.
- **A.7**: We describe the detailed algorithm steps of the TEF framework.
- **A.8**: We describe the drawbacks of TEF and future work.

## A.1 BACKBONE MODEL

To ensure a fair comparison, we downloaded the source codes of the compared methods from the authors' websites and followed the experimental settings and parameter tuning steps outlined in each paper. For the trusted fusion methods TMC, TMOA, ETMC, and RCML, as well as other non-trusted fusion methods, we meticulously tuned them on each dataset to ensure a fully fair comparison. The main comparison methods are briefly introduced below.

- **EmbraceNet** (Choi & Lee, 2019): This is a random method where each component of the fused vector comes from one view determined by a multinomial distribution. The probability value $p$ of view selection is set to $\frac{1}{|V|}$.
- **AWDR** (Yang et al., 2019): An adaptive-weighting discriminative regression method. The parameter $\lambda$ is selected from $\{10^{-3}, 10^{-2}, \cdots, 10^3\}$, and $k$ takes values from $\{1, 3, 5, \cdots, 9\}$.
- **RMAR** (Jiang et al., 2022): It adopts the $L_{2,1}$-norm loss function to learn a joint weighted projection subspace across all views, preserving correlation and diversity among views via a self-supervised weighting manner. The parameter $\lambda$ is selected from $\{10^{-3}, 10^{-2}, \cdots, 10^3\}$, and $k$ takes values from $\{1, 3, 5, \cdots, 9\}$.
- **DC-NAS** (Liang et al., 2024): It is a method that searches for the optimal multi-view fusion strategy through evolutionary neural architecture search. The divide-and-conquer approach used during the search process greatly improves efficiency while ensuring enhanced performance without sacrificing effectiveness.
- **Simple Soft Voting (SSV)** (Liang et al., 2021): It simply averages the outputs of each single view method, treating each view equally.
- **Best View (BV)** (Liang et al., 2021): A winner-take-all weighted strategy at the view level, selecting the best method among all single view methods.
- **Maximum Rule (MR)** (Liang et al., 2021): A winner-take-all weighted strategy at the example level, selecting the highest confidence score among the outputs of all single view methods.
- **TMC** (Han et al., 2021): Integrates multi-view information at an evidence level using the Dempster-Shafer theory in a learnable way.
- **TMOA** (Liu et al., 2022): Integrates multi-view information at an evidence level using opinion aggregation.
- **ETMC** (Han et al., 2023): An improved version that adds a pseudo view containing complementary information from multiple views by simply concatenating the original features.
- **RCML** (Xu et al., 2024): Fuses view-specific opinions consisting of decision results and reliability via the conflictive opinion aggregation strategy, employing a simple and effective average pooling fusion.

- **RMVC** (Yue et al., 2025b): It proposes a robust multi-view classification method based on an evidential inconsistency measure, which effectively quantifies view conflicts, designs a belief fusion strategy, and introduces gradient penalties to resist adversarial attacks.
- **TUNED** (Huang et al., 2025):It integrates feature structures, employs MRF to manage dependencies, and generates consensus evidence to address fusion uncertainties in multi-view data.

We review bilinear-based fusion (Fukui et al., 2016) and tensor-based fusion (Liu et al., 2018) as representative of advanced fusion methods. Recently, they are widely applied to many small-scale applications such as fine-grained classification (two views) (Kong & Fowlkes, 2017), visual question answering (two views) (Yu et al., 2018) and multimodal sentiment analysis (three views) (Zadeh et al., 2017).

Compared with linear methods, bilinear methods can yield a richer representation by modeling all pairwise interactions among features from different views. Two typical methods are multi-modal low-rank bilinear pooling (MLB) approach (Kim et al., 2017) and multimodal factorized bilinear pooling (MFB).

MLB (Kim et al., 2017) is to solve the dimension curse in feature fusion by using $|V| + 1$ matrix multiplication to approximate the outer product. The fusion process can be formalized as follows.

$$c = \mathrm{MLB}(v_1, v_2, \cdots, v_{|V|}) = U^{\mathrm{T}}(\bigwedge_{i=1}^{|V|} U_i^{\mathrm{T}} v_i) + b, \tag{18}$$

where $\bigwedge_{i=1}^{|V|} x_i = x_1 \circ x_2 \circ \cdots \circ x_{|V|}$; $U_i \in \mathbb{R}^{M_i \times d}$ and $c \in \mathbb{R}^m$, where $d$ and $m$ are hyper-parameters to determine the dimension of joint embeddings and the output dimension of low-rank bilinear models, respectively.

Noting that MLB could result in insufficient representation, (Yu et al., 2018) proposed an enhanced version MFB of MLB. In MFB, the features from different views are first expanded to a high-dimensional space and then integrated the expanded vectors with Hadamard product. Then sum pooling followed by the normalization layers is conducted to squeeze the high-dimensional feature into the compact output feature. The fusion process can be formalized as follows.

$$c = \mathrm{MFB}(v_1, v_2, \cdots, v_{|V|}) = \mathrm{SumPool}(\bigwedge_{i=1}^{|V|} \hat{U}_i^{\mathrm{T}} v_i, k), \tag{19}$$

where the function $\mathrm{SunPool}(x, k)$ uses a one-dimensional non-overlap window with size $k$ to do sum pooling over $x$.

Tensor-based methods model interactions among different view features by using a $|V|$-fold Cartesian product from view embeddings. Typical methods include tensor fusion network (TFN) (Zadeh et al., 2017), low-rank multimodal fusion method (LMF) (Liu et al., 2018) and polynomial tensor pooling (PTP) (Hou et al., 2019).

TFN (Zadeh et al., 2017) introduce a tensor fusion layer. Given $|V|$ view vectors $\{v_i \in \mathbb{R}^{m_i}\}_{i=1}^{|V|}$, they are fused as follows:

$$\mathcal{Z} = \begin{bmatrix} v_1 \\ 1 \end{bmatrix} \otimes \begin{bmatrix} v_2 \\ 1 \end{bmatrix} \otimes \cdots \otimes \begin{bmatrix} v_{|V|} \\ 1 \end{bmatrix}, \tag{20}$$

where $\otimes$ is the Kronecker product operator, namely outer product, and $\mathcal{Z} \in \mathbb{R}^{(m_1+1) \times (m_2+1) \times \cdots \times (m_{|V|}+1)}$ is an order-$|V|$ tensor.

Then the tensor $\mathcal{Z}$ is mapped into a vector by a linear transformation as follows:

$$c = \mathcal{WZ} + b, \tag{21}$$

where $\mathcal{W} \in \mathbb{R}^{(m_1+1) \times (m_2+1) \times \cdots \times (m_{|V|}+1) \times m}$, $b \in \mathbb{R}^m$ and $c \in \mathbb{R}^m$.

Noting that in TFN the number of parameters to learn in the weight tensor $\mathcal{W}$ will increase exponentially with the number of views, Liu et al. (Liu et al., 2018) proposed a low-rank multimodal fusion method (LMF). The fusion process can be formalized as follows.

$$c = \text{LMF}(v_1, v_2, \cdots, v_{|V|}) = \bigwedge_{i=1}^{|V|} \sum_{j=1}^{r} (W_i^{(j)} \begin{bmatrix} v_i \\ 1 \end{bmatrix}), \tag{22}$$

where $r$ denotes the rank or the decomposition factors of a tensor; $W_i^{(j)} \in \mathbb{R}^{m \times (m_i+1)}$ and $c \in \mathbb{R}^m$.

Noting that tensor-based multimodal fusion methods simply fuse features all at once, ignoring the complex local intercorrelations, Hou et al. (Hou et al., 2019) proposed a polynomial tensor pooling (PTP) by considering high-order moments, followed by a tensorized fully connected layer. Given $|V|$ view vectors $\{\boldsymbol{v}_i \in \mathbb{R}^{m_i}\}_{i=1}^{|V|}$, they are fused as follows:

$$\mathcal{F} = \underbrace{\boldsymbol{f} \otimes \boldsymbol{f} \otimes \cdots \otimes \boldsymbol{f}}_{P-\text{order}},$$

where $\boldsymbol{f} = [1, \boldsymbol{v}_1, \boldsymbol{v}_2 \cdots \boldsymbol{v}_{|V|}] \in \mathbb{R}^a$ and $\mathcal{F} \in \mathbb{R}^{a \times a \times, \cdots \times a}$; $a = \sum_{i=1}^{|V|} m_i + 1$.

Then the tensor $\mathcal{F}$ is mapped into a vector by a linear transformation as follows:

$$\boldsymbol{c} = \text{PTP}(v_1, v_2, \cdots, v_{|V|}) = \mathcal{W}\mathcal{F} = (\sum_{j=1}^{r} \bigotimes_{i=1}^{|V|} \mathbf{w}_i^{(j)})\mathcal{F}, \tag{23}$$

where $\mathcal{W} \in \mathbb{R}^{a \times a \times \cdots \times a \times m}$ and $c \in \mathbb{R}^m$.

Despite achieving promising performance, the manual multi-view fusion might be sub-optimal for message propagation between views. Moreover, the dimensionality of the fused vector that some of them produce increases exponentially as the number of view increases and are unsuitable for many-view data. Hence, we adopt an evolutionary strategy for automatically discovering well-suitable many-view fusion scheme. For a good computational efficiency in fusing many-view data, the evolved fusion scheme only includes the basic fusion operators.

## A.2 FIVE BASIC FUSION OPERATORS

To better illustrate the fusion process of multiple fusion operators, we define $x_i$ as the vector feature, $n$ represents the quantity of vector features to be fused, where the superscript indicates the index of the fused vector feature. In this context, the fusion operator set $F$ encompasses the following operations (Liang et al., 2021; Fu et al., 2025).

*Concatenation*: The information from vector features is fused as follows:

$$o(x_i) = [x_i^1, x_i^2, \cdots, x_i^{|n|}], \tag{24}$$

where $[\cdot, \cdot]$ is the concatenation operator.

Element-wise fusion operators require that the dimensions of input vectors are the same, hence different vector features need to be mapped into the same dimension space by a linear function before fusing. This can be achieved using a fully-connected layer (FC) without any activation function.

*Addition*: The information from vector features is fused as follows:

$$o(x_i) = \text{FC}(x_i^1) + \text{FC}(x_i^2) + \cdots + \text{FC}(x_i^{|n|}). \tag{25}$$

*Multiplication*: The information from vector features is fused as follows:

$$o(x_i) = \text{FC}(x_i^1) \circ \text{FC}(x_i^2) \circ \cdots \circ \text{FC}(x_i^{|n|}), \tag{26}$$

where $\circ$ denotes Hadamard product, namely element-wise multiplication.

*Max*: The information from vector features is fused as follows:

$$o(x_i) = \max(\text{FC}(x_i^1), \text{FC}(x_i^2) \cdots, \text{FC}(x_i^{|n|})), \tag{27}$$

where max is element-wise max, also called max-pooling.

*Average*: The information from vector features is fused as follows:

$$o(x_i) = \frac{1}{|n|}(\text{FC}(x_i^1) + \text{FC}(x_i^2) + \cdots + \text{FC}(x_i^{|n|}), \tag{28}$$

where $+$ denotes element-wise addition, also called average-pooling.

Table 7: Statistic information of datasets

| View | AWA | NUS | Reuters | MVoxCeleb | YouTube Faces |
|------|-----|-----|---------|-----------|---------------|
| $V_1$ | CH(2,688) | CH(64) | EN(21,531) | ecapa(192) | $V_1$(64) |
| $V_2$ | LSS(2,000) | CM(225) | FR(24,893) | resnet(512) | $V_2$(512) |
| $V_3$ | PHOG(252) | CORR(144) | GR(34,279) | fbank(160) | $V_3$(64) |
| $V_4$ | SIFT(2,000) | EDH(73) | IT(15,506) | mfcc(80) | $V_4$(647) |
| $V_5$ | SURF(2,000) | WT(128) | SP(11,547) | spec(514) | $V_5$(838) |
| $V_6$ | rgSIFT(2,000) | BoW(500) | - | - | - |
| $V_7$ | ResNet101(2,048) | Tags1k(1,000) | - | - | - |
| #Sample | 30,475 | 23,438 | 111,740 | 153,516 | 101,499 |
| #Label | 50 | 10 | 6 | 1,251 | 31 |

## A.3 DATASETS DETAILS

We describe the datasets used in the experiments in detail and summarize the datasets in Table 7.

- **Animals with Attributes** (AWA) (Lampert et al., 2014): This dataset includes 30,475 images of 50 animal subjects with seven views. For each image, six types of low-level features are extracted: RGB color histogram (CH), local self-similarity histograms (LSS), PyramidHOG (PHOG), SIFT, rgSIFT, SURF, and one deep feature (ILSVRC-pretrained ResNet101).

- **NUS-WIDE-128** (NUS) (Tang et al., 2017): This dataset contains 43,800 single-label images from 128 categories. For each image, six types of image features are extracted: color histogram (CH), color correlogram (CORR), edge direction histogram (EDH), wavelet texture (WT), block-wise color moments (CM), and bag of words based on SIFT descriptions (BoW), along with one text feature. The dataset is an extension of the NUS-WIDE dataset (Chua et al., 2009). In our experiments, we use a subset consisting of 23,438 images from 10 categories, including *animal*, *architecture*, *art*, *flowers*, *food*, *man*, *person*, *sky*, *toy*, and *water*. Each image in this subset is related to one label, and each category includes at least 1,500 images.

- **Reuters** (Amini et al., 2009): This is a multilingual multi-view dataset where each document is described by five different languages: English, French, German, Spanish, and Italian. To make the model efficiently work on this data, we reduce the dimensions of all views to 1,000 using PCA and add the Gaussian noise to all views or three views, resulting in two versions named Reuters5 and Reuters3, respectively.

- **MVoxCeleb**: This is a multi-view audio classification dataset constructed from the Vox-Celeb dataset (Nagrani et al., 2020). Specifically, we extract each audio sample five view features extracted: two deep features (ecapa and resnet) and three traditional features (fbank, mfcc, and spec) and add the Gaussian noise to the ecapa and resnet mfcc to study the effect.

- **YouTube-Faces** (Wang et al., 2022): This dataset includes 3,425 videos of 1,595 different people downloaded from YouTube. Similar to other datasets, we use a subset consisting of 101,499 frames of 31 subjects, with the same five features extracted.

We additionally introduced seven standard datasets commonly used in TMVC methods to compare with the TEF method and further demonstrate its performance advantages. These datasets were not used in the main text primarily due to their smaller number of views and samples, as the TEF method can complete experiments quickly using evolutionary search within a short time. To better highlight the advantages of TEF, we selected six more challenging datasets for demonstration, but we have also included the comparison results for these seven datasets in the appendix.

- **PIE**: It contains 680 instances, divided into 68 categories, with three types of features extracted: intensity, LBP, and Gabor.

- **HandWritten**: It has 2000 handwritten digit instances, represented by six sets of features.

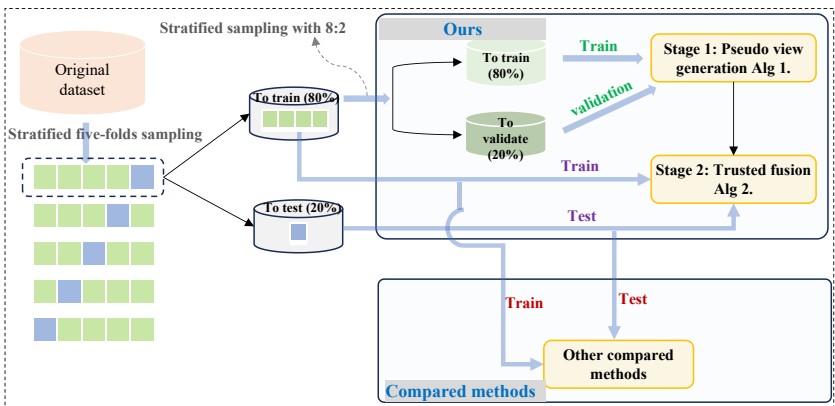

Figure 4: Details of the dataset partitioning for the training set, test set, and validation set in the experimental setup.

- **Scene15**: It consists of 4485 indoor and outdoor scene images, with GIST, PHOG, and LBP features extracted.

- **Caltech101**: It contains 101 categories of images, where the top 10 categories were selected, and deep features were extracted using DECAF and VGG19.

- **CUB**: It contains 11,788 bird instances, and image and text features were extracted from the top 10 categories.

- **Animal**: It includes 50 categories with a total of 10,158 images, where deep features were extracted using DECAF and VGG19.

- **NUS-WIDE-OBJECT**: It consists of 31 categories with 30,000 images, described using five different perspectives.

### A.4 IMPLEMENTATION DETAILS

In the experiments, to reduce the randomness introduced by data partitioning and network initialization, we followed the strategy proposed in the literature (Shi et al., 2022). As shown in the Figure 4, we first performed 5-fold cross-validation on the original dataset, dividing the data into training and testing sets in an 8:2 ratio. In the first phase of the evolutionary search pseudo-view generation framework, the training set was further split into a training set and a validation set to prevent data leakage. In the second phase, we continued to maintain a consistent partitioning strategy with all comparison methods. The reported data represents the average performance and standard deviation from five data partitions. To ensure fairness in comparison, we conducted meticulous hyperparameter tuning for each method to optimize its performance. In multi-view fusion, unifying dimensions is a key step; therefore, we standardized the dimensions of all methods to 128.

Our computational environment was Ubuntu 16.04.4, with 512 GB DDR4 RDIMM, 2x 40-core Intel Xeon CPU E5-2698 v4 @ 2.20 GHz, and NVIDIA Tesla P100 (16 GB GPU memory). TEF involves parameters across two stages: First Stage - DNN Training: All DNN models were trained using the Adam algorithm. The learning rate was set to 0.001, the exponential decay rate for the first moment estimate was 0.9, and for the second moment estimate was 0.999. Each network was trained for 100 epochs. To avoid overfitting, the training process was stopped if the neural network model performance did not improve after 10 epochs. Evolutionary Algorithm Parameters: To effectively utilize GPU resources, the population size was set as a multiple of the number of GPUs. Using 7 NVIDIA Tesla P100 GPUs, the population size was set to 28. Following (Shi et al., 2022), the number of generations was set to 20, with crossover and mutation probabilities set to 0.9 and 0.2, respectively. It is noteworthy that the same set of parameters was used for all six datasets. Our main goal was to obtain a fusion architecture to generate high-quality pseudo views in the first stage. The detailed hyperparameters used in the second stage of TEF for the six datasets are shown in Table 8.

Table 8: Detailed hyperparameters for TEF running on various datasets.

|  | AWA | NUS | Reuters5 | Reuters3 | VoxCeleb | YoutubeFace |
|---|---|---|---|---|---|---|
| Learning rate | 1e-3 | 1e-3 | 1e-4 | 1e-4 | 1e-4 | 1e-4 |
| Weight decay | 1e-5 | 1e-5 | 1e-5 | 1e-3 | 1e-4 | 1e-5 |
| Batch size | 256 | 64 | 128 | 64 | 256 | 256 |
| Optimizer | Adam | Adam | Adam | Adam | Adam | Adam |
| Maximum Epoch | 500 | 500 | 500 | 500 | 500 | 500 |
| Fusion dimension | 128 | 128 | 128 | 128 | 128 | 128 |
| Lr-patience | 20 | 20 | 10 | 10 | 50 | 10 |
| Lr-factor | 0.3 | 0.3 | 0.3 | 0.3 | 0.3 | 0.3 |
| Seed | 1 | 1 | 1 | 1 | 1 | 1 |

Table 9: Comparison results on the impact of view enhancement position on TEF (mean ± standard deviation), with the best performance highlighted in boldface.

| | Accuracy | | | | | |
|---|---|---|---|---|---|---|
| Methods | AWA | NUS | Reuters5 | Reuters3 | VoxCeleb | YoutubeFace |
| $TEF^1$ | 88.59±0.25 | 72.73±0.30 | 79.60±0.56 | 84.23±0.35 | 73.13±0.15 | 71.18±2.27 |
| $TEF^2$ | 91.60±0.20 | 73.71±0.48 | 79.93±0.54 | 85.33±0.41 | 91.44±0.14 | 76.68±1.44 |
| $TEF^3$ | 93.16±1.21 | 75.06±0.66 | 81.14±0.63 | 86.45±0.18 | 92.06±0.14 | 85.35±0.62 |
| $TEF$ | **93.26±1.25** | **75.12 ± 0.57** | **82.26±0.23** | **86.49±0.10** | **92.41±0.12** | **86.02±0.41** |

## A.5 ADDITIONAL RESULTS

In this section, we provide additional ablation studies and experimental results on the TEF framework to facilitate a comprehensive understanding of the TEF architecture. They include:

1. Impact analysis of fusion dimensions on TEF.

2. Impact analysis of the position where views are enhanced on TEF.

3. Impact analysis of layer where features are extracted to use view enhancement on TEF.

4. Impact analysis of different forms of encoding individual.

These experimental results help to better understand the key factors of the TEF architecture and its performance in different scenarios.

**Impact analysis of fusion dimensions on TEF.** We conducted a discussion on the fusion dimensions on the AWA dataset, examining the effects of different dimensions: 32, 64, 128, 256, and 512. Fig. 5 shows the performance of five metrics at these dimensions. It can be seen that as the fusion dimension increases, the performance of TEF improves significantly. For example, when the fusion dimension is 32, the accuracy is 92.41%, and it increases to 93.51% when the fusion dimension reaches 256. This improvement is because higher dimensions bring more learnable parameters, thereby enhancing the model's learning capacity. As the dimensions increase, the model's learning capacity tends to stabilize, with changes between 256 and 512 not being significant. Considering all factors, we chose 128 as the fusion dimension to balance performance and model size. It is worth noting that to ensure a fair comparison, the fusion dimension for all methods compared in this paper is set to 128.

**Impact analysis of the position where views are enhanced on TEF.** This part aims to investigate impact of enhancement position on TEF by comparing TEF with its three modified versions. Specifically, $TEF^1$ denotes that the view enhancement is removed from the TEF; $TEF^2$ denotes that only decision output of the pseudo view network is used to pseudo view; $TEF^3$ denotes the decision output of the pseudo view network is concatenated the fusion layer of the pseudo view network. From Table 9, it is clear that both the use of feature concatenation and the position where the features are concatenated impact the performance of the TEF architecture. Compared with $TEF^1$, $TEF$ demonstrates a significant performance advantage across all datasets, such as a 9.44% improvement on the

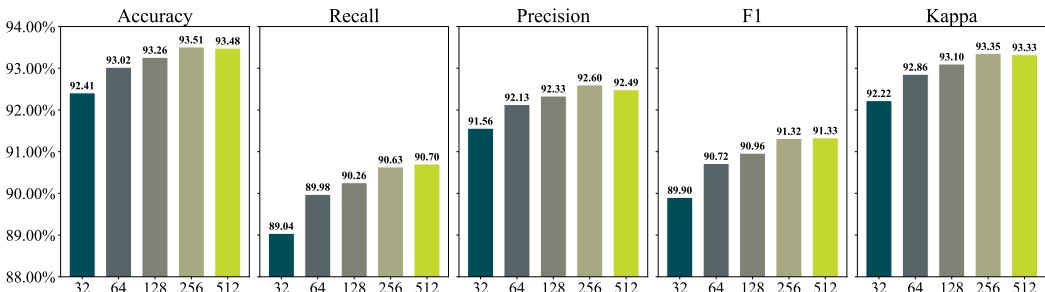

Figure 5: The impact of different fusion dimensions on the TEF architecture.

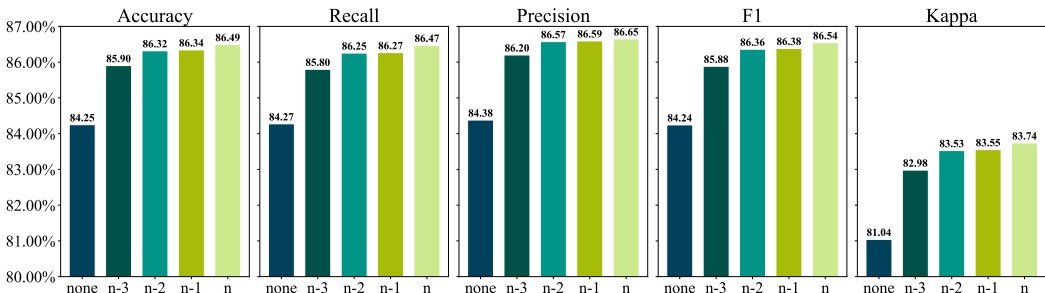

Figure 6: The impact of different feature layers on TEF's ability to solve the imbalance in views.

YoutubeFace dataset. Compared to $TEF^3$, TEF shows that the strategy of concatenating the decision output of the pseudo view network into the original views is generally superior to concatenating them after view fusion but before entering the hierarchical stage. This indicates that concatenation at the original view stage better preserves and utilizes multi-view information, thereby enhancing the performance of the TEF architecture.

**Impact analysis of layer where features are extracted to use view enhancement on TEF.** We explored the effects of concatenating features at different layers to enhance the original views on the Reuters3 dataset. Specifically, we concatenated features from the last four layers of the fusion architecture to evaluate the final effect of feature enhancement in TEF. Assuming the fusion architecture has n layers, we extracted features from the nth layer, $(n-1)$th layer, $(n-2)$th layer, and $(n-3)$th layer, as well as a baseline without using any features. The experimental results are shown in Fig. 6. The closer to the last layer, the better the performance. For example, in terms of accuracy, concatenating features from the nth layer is 0.59% better than concatenating features from the $(n-3)$th layer. This is because the deepest layers of the fusion architecture often contain the most complex and advanced representations of the model. These features tend to capture deeper patterns and more semantic information in the data. Using post-fusion multi-view representations to enhance the original views helps improve the representational capacity of the views, preventing information loss during fusion. This approach contributes to achieving better results in multi-view fusion architecture and helps mitigate the multi-view imbalance problem to some extent.

**Impact analysis of different forms of encoding individual.** The individual can be encoded in different forms. Here, we further offer its a binary tree form where the views are used as the leaf nodes of the tree while the fusion operators are used as the non-leaf nodes. Similar to its tuple form, if the binary tree contains $k$ views, then it must contain $k-1$ fusion operators. From Fig. 7, it can be observed that the binary tree encoding strategy performs slightly better than the tuple form. However, for simplicity and generality of the architecture, we initially chose the tuple form. Of course, TEF can naturally switch between both of them.

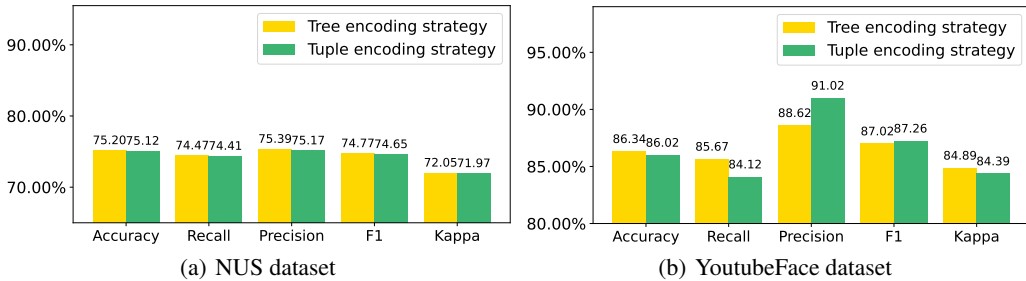

Figure 7: A comparative analysis of different encoding methods across five evaluation metrics.

Table 10: Comparison on benchmark datasets used by other TMVC methods. "†" and "*" indicate that the results are reported in the work (Xu et al., 2024) and (Huang et al., 2025), respectively.

| Method | PIE | HandWritten | Scene15 | Caltech101 | CUB | Animal | NUS |
|---|---|---|---|---|---|---|---|
| EDL (NeurIPS18) | 86.25±0.89* | 96.90±0.16* | 52.76±0.54* | 73.35±1.73* | 86.22±0.36* | 84.30±1.76* | 22.33±0.64* |
| DCCAE (ICML15) | 81.96±1.04† | 95.45±0.35† | 74.62±1.52† | 89.56±0.41† | 85.93±1.36† | 82.72±1.38* | 35.75±0.48* |
| CPM-Nets (ICML19) | 88.53±1.23† | 94.55±1.36† | 67.29±1.01† | 90.35±2.12† | 89.32±0.38† | 87.40±1.12* | 35.37±1.05* |
| DUA-Nets (AAAI21) | 90.56±0.47† | 98.10±0.32† | 68.23±0.11† | 93.43±0.04† | 80.13±1.67† | 78.65±0.55* | 39.38±0.34* |
| TMC (ICLR22) | 91.85±0.23† | 98.51±0.13† | 67.71±0.30† | 92.80±0.50† | 90.57±2.96† | 79.31±0.43* | 35.18±1.55* |
| TMOA (AAAI22) | 92.33±0.36† | 99.25±0.45† | 75.57±0.02† | 94.63±0.04† | 95.43±0.20† | 87.05±0.28* | 34.39±0.44* |
| RCML (AAAI24) | 94.71±0.02† | 99.40±0.00† | 76.19±0.12† | 95.36±0.38† | 94.50±1.13† | 84.01±0.63* | 34.04±0.27* |
| RMVC (IF25) | 91.18±0.24* | 98.51±0.04* | 73.05±0.24* | 88.73±0.60* | 93.18±0.47* | 87.67±0.17* | 34.68±0.32* |
| TUNED (AAAI25) | 96.83±0.01* | 99.20±0.23* | 75.89±0.70* | **97.38±0.45*** | **99.00±0.33*** | 89.52±0.13* | 37.46±0.25* |
| **TEF (Ours)** | **97.57±0.78** | **99.64±0.13** | **78.00±0.48** | 96.04±0.32 | 95.92±0.62 | **90.18±0.08** | **47.52±0.30** |

Table 11: Comparison on benchmark conflicting datasets used by other TMVC methods. "†" and "*" indicate that the results are reported in the work (Xu et al., 2024) and (Huang et al., 2025), respectively.

| Method | PIE | HandWritten | Scene15 | Caltech-101 | CUB | Animal | NUS |
|---|---|---|---|---|---|---|---|
| EDL (NeurIPS18) | 21.76±0.67* | 57.25±0.49* | 14.28±0.24* | 55.74±0.12* | 53.75±0.42* | 37.10±0.27* | 18.07±0.28* |
| DCCAE (ICML15) | 26.89±1.10† | 82.85±0.38† | 25.97±2.86† | 60.90±2.32† | 63.57±1.28† | 34.02±0.11* | 32.12±0.52* |
| CPM-Nets (ICML19) | 53.19±1.17† | 83.34±1.07† | 29.63±1.12† | 66.54±2.89† | 68.82±0.17† | 64.83±0.35* | 29.20±0.81* |
| DUA-Nets (AAAI21) | 56.45±1.75† | 87.16±0.34† | 26.18±3.10† | 75.19±2.34† | 60.53±1.17† | 62.46±1.12* | 31.82±0.43* |
| TMC (ICLR22) | 61.65±1.03† | 92.76±0.15† | 42.27±1.61† | 90.16±2.40† | 73.37±2.16† | 64.85±1.19* | 33.76±2.16* |
| TMOA (AAAI22) | 68.16±0.34† | 93.05±0.45† | 48.42±1.02† | 90.63±2.35† | 74.43±0.36† | 64.20±0.15* | 32.44±0.26* |
| RCML (AAAI24) | 84.00±0.14† | 94.40±0.05† | 56.97±0.52† | 92.36±1.48† | 76.50±1.15† | 67.67±0.81* | 31.91±0.22* |
| RMVC (IF25) | 76.47±3.43* | 94.75±0.75* | 49.83±2.23* | 80.56±0.71* | 72.78±0.42* | 66.00±0.59* | 24.63±1.19* |
| TUNED (AAAI25) | 86.02±0.19* | 96.75±0.55* | 67.22±0.58* | 93.22±0.41* | 76.67±0.38* | **90.90±0.18*** | 34.09±0.14* |
| **TEF (Ours)** | **86.76±0.49** | **98.70±0.31** | **72.27±0.43** | **93.42±0.58** | **77.41±0.47** | 76.10±0.12 | **45.84±0.31** |

## A.6 COMPARISON ON BENCHMARK DATASETS USED BY OTHER TMVC METHODS

To fully demonstrate the advantages of our method and further verify its fairness, we also conducted evaluations on seven benchmark datasets commonly used by other TMVC methods (Wang et al., 2015; Zhang et al., 2019; Geng et al., 2021; Yue et al., 2025a; Huang et al., 2025). Using the same data partitioning strategy as RCML (Xu et al., 2024), we repeated the experiments ten times randomly and calculated the average results and standard deviations. The results shown in Table 10 indicate that our method exhibits significant advantages across all seven datasets.

Moreover, we show TEF's effectiveness in handling noisy or uncertain data by conducting rigorous tests on seven conflicting datasets, structured according to the method described by RCML (Xu et al., 2024). TEF underwent ten iterations of testing on each dataset to ensure statistical robustness, with both mean values and standard deviations reported. In Table 11, these results confirm TEF's effectiveness in managing datasets with inherent noise or uncertainty. As shown, TEF consistently delivers superior performance, even under challenging conditions involving noisy or uncertain data. Notably, on the NUS dataset, TEF achieved a performance that was 11.75% higher than the second-best method, and on the Scene15 dataset, it surpassed the next best by 5.05%.

## A.7 THE TEF FRAMEWORK ALGORITHM

The trusted multi-view classification enhancement through evolutionary view fusion (TEF) approach consists of two stages: (1) Using neural architecture search to obtain the optimal pseudo-view generation model, and extracting subsequent decision features for feature enhancement in the second stage, as detailed in Alg. 1. (2) Fusion view-guided trusted fusion through evidence fusion, as referenced in Alg. 2. The detailed evolutionary operations of Alg. 1 are described as follows:

**Population Initialization**: Randomly generate $N$ chromosome vectors as the initial population, where each chromosome $p$ has a different $k$-value.

**Fitness evaluation**: Decode chromosome $p$ into a deep fusion neural network model, train it on a multi-view dataset, and evaluate its fitness based on classification accuracy on the validation set.

**Crossover operation**: Perform crossover between two chromosomes $p_1$ and $p_2$ to generate new chromosome vectors $p_1^0$ and $p_2^0$.

**Mutation operation**: Randomly select an individual $p$ and, with a certain probability, replace its view $v_i$ or fusion operator $f_j$ to achieve mutation.

---

**Algorithm 1** Evolutionary NAS method

1: **Input:**
2: $N$: population size;
3: $T$: maximal generation number;
4: $D = (X, Y)$: training dataset;
5: $\hat{D} = (\hat{X}, \hat{Y})$: validation dataset;
6: $F$: a set of basic fusion operators;
7: **Output:** Optimal fusion architecture and decision output features.
8: Generate an initial population $P_0$;
9: Evaluate the fitness of each chromosome vector in $P_0$;
10: **for** $t = 1$ to $T$ **do**
11:     Generate offspring $Q_t$ using the crossover operator;
12:     Conduct mutation on each chromosome in $Q_t$;
13:     Evaluate the fitness of each chromosome in $Q_t$;
14:     Select next generation population $P_{t+1}$ from $Q_t \cup P_t$ using a selection operator;
15: **end for**
16: $p_{\text{best}} \leftarrow$ Select the chromosome with the best fitness from $P_T$.
17: **return** The fusion network corresponding to $p_{\text{best}}$ and obtain the decision output features of this network.

---

## A.8 LIMITATIONS AND FUTURE WORKS

Even though the proposed method achieves excellent performance, it still has some potential limitations. (1) Like other trustworthy fusion methods, we did not adjust the fusion order of the trustworthy fusion process. Specifically, when we form opinions $M_1$, $M_2$, ..., $M_{n+1}$ from $n$ views and an additional $(n + 1)$th pseudo-view, their aggregation order is crucial as different orders can lead to different performance impacts. Therefore, in future work, we will conduct a theoretical analysis of this phenomenon and propose solutions. (2) Additionally, not all opinions need to be aggregated. For example, some views may generate opinions that are entirely noise or significantly degrade the fusion results. We need to exclude these invalid opinions to ensure the effectiveness of the fusion. In future work, we envision that an adaptive method for effectively selecting and aggregating these opinions to ensure the effectiveness of the entire framework and improve performance will be a promising research direction. We consider these as important future works.

---

**Algorithm 2** Algorithm for trusted multi-view classification

---

1: **Training**
2: **Input:** Multi-view dataset: $\mathcal{D} = \left\{ \{\mathbf{X}_n^m\}_{m=1}^V \cup \mathbf{X}_n^{sp}, y_n \right\}_{n=1}^N$
3: **Initialize:** Initialize the parameters of the neural network.
4: **while** not converged **do**
5:     **for** $m = 1$ to $V$ **do**
6:         $\text{Dir}(\boldsymbol{\mu}^m \mid \mathbf{x}^m) \leftarrow$ variational network output;
7:         Subjective opinion $\mathbf{M}^m \leftarrow \text{Dir}(\boldsymbol{\mu}^m \mid \mathbf{x}^m)$;
8:     **end for**
9:     $\text{Dir}(\boldsymbol{\mu}^{sp} \mid \mathbf{x}^{sp}) \leftarrow$ variational network output;
10:     Subjective opinion $\mathbf{M}^{sp} \leftarrow \text{Dir}(\boldsymbol{\mu}^{sp} \mid \mathbf{x}^{sp})$;
11:     Obtain joint opinion $\mathbf{M}$ with Eq. (11);
12:     Obtain $\text{Dir}(\boldsymbol{\mu} \mid \boldsymbol{\alpha})$ with Eq. (13);
13:     Obtain the overall loss by updating $\boldsymbol{\alpha}$ and $\{\alpha^v\}_{v=1}^V$ in Eq. (14);
14:     Maximize Eq. (14) and update the networks with gradient descent;
15: **end while**
16: **Output:** networks parameters.
17: **Test**
18: Calculate the joint belief and the uncertainty masses.

---

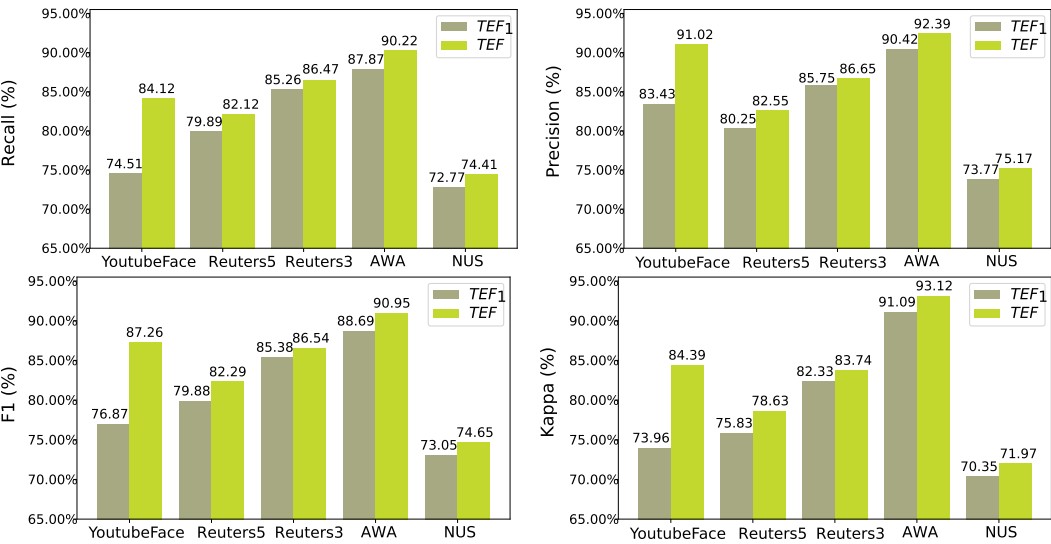

Figure 8: Changes in TEF performance before and after addressing the view imbalance issue under different metrics (Recall, Precision, F1, and Kappa).

Table 12: Comparison results of the other four evaluation metrics with SOTA algorithms (mean $\pm$ std), with the best performance highlighted in boldface.

| | | | Recall | | | |
|---|---|---|---|---|---|---|
| Methods | AWA | NUS | Reuters5 | Reuters3 | VoxCeleb | YoutubeFace |
| TMC (ICLR22) | 84.47±0.54 | 71.70±0.43 | 79.60±0.56 | 84.19±0.29 | 64.06±0.12 | 68.50±2.77 |
| TMOA (AAAI22) | 83.62±0.91 | 71.81±0.49 | 78.85±0.30 | 84.25±0.30 | 81.54±0.26 | 82.63±0.39 |
| ETMC (TPAMI23) | 83.52±0.51 | 72.35±0.79 | 79.74±0.52 | 83.51±0.51 | 85.85±0.22 | 81.51±0.26 |
| RCML (AAAI24) | 84.34±0.88 | 71.55±0.68 | 81.52±0.16 | 85.81±0.27 | 74.53±0.46 | 80.59±0.18 |
| EmbraceNet | 80.04±0.59 | 72.04±0.34 | 79.85±0.26 | 83.46±0.21 | 78.36±0.34 | 80.65±1.13 |
| AWDR (PR19) | 86.86±0.20 | 71.87±0.62 | 79.59±0.23 | 83.30±0.29 | 87.26±0.13 | 83.57±0.30 |
| RMAR (INS22) | 87.08±0.42 | 71.92±0.65 | 79.73±0.23 | 83.45±0.23 | 87.95±0.11 | 83.35±0.27 |
| DC-NAS (AAAI24) | 87.82±0.34 | 73.75±0.71 | 81.23±0.21 | 85.77±0.09 | **90.14±0.10** | **84.69±0.46** |
| **TEF (Ours)** | **90.26±1.41** | **74.41±0.60** | **82.12±0.17** | **86.47±0.10** | 89.81±0.20 | 84.11±0.84 |

| | | | Precision | | | |
|---|---|---|---|---|---|---|
| Methods | AWA | NUS | Reuters5 | Reuters3 | VoxCeleb | YoutubeFace |
| TMC (ICLR22) | 87.76±0.40 | 72.71±0.22 | 79.86±0.46 | 84.43±0.49 | 73.26±0.34 | 82.53±2.01 |
| TMOA (AAAI22) | 88.15±0.62 | 72.73±0.53 | 79.89±0.72 | 84.40±0.23 | 84.38±0.30 | 87.59±0.28 |
| ETMC (TPAMI23) | 87.68±0.63 | 72.39±0.64 | 79.99±0.33 | 84.38±0.37 | 87.28±0.15 | 83.40±2.33 |
| RCML (AAAI24) | 86.27±1.22 | 73.05±0.26 | 81.52±0.18 | 85.81±0.27 | 74.21±0.46 | 84.34±0.88 |
| EmbraceNet | 82.14±0.57 | 71.73±0.32 | 80.42±0.25 | 83.77±0.34 | 80.95±0.46 | 83.71±1.10 |
| AWDR (PR19) | 89.32±0.33 | 72.71±0.61 | 79.87±0.30 | 83.49±0.34 | 91.83±0.11 | 89.94±0.32 |
| RMAR (INS22) | 89.41±0.38 | 72.82±0.64 | 80.12±0.27 | 83.70±0.28 | 92.19±0.06 | 90.64±0.08 |
| DC-NAS (AAAI24) | 88.99±0.27 | 73.82±0.45 | 81.53±0.32 | 86.07±0.21 | 91.73±0.10 | 87.14±0.88 |
| **TEF (Ours)** | **92.33±1.23** | **75.17±0.55** | **82.55±0.32** | **86.65±0.18** | **93.01±0.19** | **91.02±1.02** |

| | | | F1 | | | |
|---|---|---|---|---|---|---|
| Methods | AWA | NUS | Reuters5 | Reuters3 | VoxCeleb | YoutubeFace |
| TMC (ICLR22) | 85.28±0.54 | 71.84±0.31 | 79.52±0.57 | 84.22±0.38 | 65.22±0.09 | 71.92±2.06 |
| TMOA (AAAI22) | 83.65±0.86 | 72.02±0.49 | 79.14±0.49 | 84.24±0.24 | 82.02±0.33 | 84.85±0.25 |
| ETMC (TPAMI23) | 84.60±0.49 | 72.19±0.68 | 79.72±0.40 | 84.24±0.42 | 86.03±0.20 | 80.97±1.48 |
| RCML (AAAI24) | 84.82±1.05 | 72.01±0.52 | 81.35±0.16 | 85.89±0.28 | 75.87±0.35 | 82.30±0.15 |
| BV | 85.94±0.50 | 68.64±0.63 | 80.61±0.24 | 83.99±0.11 | 58.34±0.23 | 82.49±0.25 |
| SSV | 78.82±1.42 | 62.13±0.69 | 79.49±0.42 | 84.75±0.21 | 81.75±0.23 | 86.55±0.27 |
| MR | 84.14±0.73 | 62.96±0.93 | 78.11±0.46 | 84.16±0.19 | 75.88±0.30 | 85.03±0.29 |
| EmbraceNet | 80.60±0.62 | 71.78±0.36 | 80.07±0.22 | 83.59±0.25 | 78.64±0.41 | 81.61±0.99 |
| AWDR(PR19) | 87.72±0.21 | 72.16±0.62 | 79.71±0.27 | 83.37±0.30 | 88.57±0.13 | 86.51±0.12 |
| RMAR(INS22) | 87.92±0.30 | 72.21±0.65 | 79.90±0.25 | 83.54±0.24 | 89.20±0.10 | 86.69±0.17 |
| DC-NAS (AAAI24) | 88.18±0.27 | 73.60±0.52 | 81.35±0.26 | 85.90±0.13 | 90.59±0.05 | 85.81±0.19 |
| **TEF (Ours)** | **90.96±1.30** | **74.65±0.56** | **82.29±0.24** | **86.54±0.10** | **90.87±0.12** | **87.26±0.38** |

| | | | Kappa | | | |
|---|---|---|---|---|---|---|
| Methods | AWA | NUS | Reuters5 | Reuters3 | VoxCeleb | YoutubeFace |
| TMC (ICLR22) | 88.31±0.26 | 69.22±0.35 | 75.44±0.68 | 81.01±0.42 | 73.10±0.15 | 67.68±2.46 |
| TMOA (AAAI22) | 88.91±0.31 | 69.12±0.53 | 74.82±0.50 | 80.97±0.33 | 84.55±0.42 | 82.60±0.29 |
| ETMC (TPAMI23) | 87.95±0.17 | 69.64±0.77 | 75.67±0.51 | 81.04±0.51 | 88.69±0.15 | 77.63±1.85 |
| RCML (AAAI24) | 88.79±0.21 | 69.00±0.64 | 77.59±0.21 | 82.99±0.34 | 80.85±0.41 | 79.97±0.22 |
| EmbraceNet | 84.60±0.24 | 68.99±0.44 | 75.99±0.25 | 80.23±0.30 | 81.72±0.34 | 78.93±1.12 |
| AWDR (PR19) | 90.23±0.06 | 68.96±0.75 | 75.54±0.32 | 79.91±0.38 | 91.07±0.09 | 83.40±0.18 |
| RMAR (INS22) | 90.41±0.13 | 69.04±0.78 | 75.72±0.29 | 80.11±0.30 | 91.53±0.11 | 83.47±0.21 |
| DC-NAS (AAAI24) | 90.43±0.15 | 71.14±0.66 | 77.54±0.33 | 82.97±0.17 | 92.19±0.08 | 83.74±0.16 |
| **TEF (Ours)** | **93.10±1.29** | **71.97±0.64** | **78.63±0.28** | **83.74±0.12** | **92.40±0.12** | **84.39±0.49** |

Table 13: The comparison results between TEF and five basic fusion operators as well as five advanced fusion operators, with the best performance highlighted in bold. The table shows four metrics: Recall, Precision, F1-score, and Kappa.

| Methods | AWA | NUS | Reuters5 | Reuters3 | VoxCeleb | YoutubeFace |
|---|---|---|---|---|---|---|
| Basic fusion operators (Recall) | | | | | | |
| Add | 85.16±0.18 | 72.26±0.74 | 79.53±0.22 | 83.34±0.28 | 84.70±0.56 | 82.43±0.26 |
| Mul | 82.60±0.45 | 63.38±0.76 | 76.52±0.47 | 81.63±0.80 | 67.70±1.03 | 83.30±1.01 |
| Cat | 84.62±0.30 | 72.03±0.39 | 79.82±0.24 | 83.59±0.21 | 84.94±0.20 | 83.40±0.47 |
| Max | 85.55±0.32 | 70.80±0.37 | 79.87±0.23 | 83.94±0.21 | 78.02±0.33 | 80.75±0.70 |
| Avg | 85.37±0.44 | 72.54±0.55 | 79.54±0.26 | 83.46±0.32 | 84.54±0.45 | 81.38±0.99 |
| Advanced fusion operators (Recall) | | | | | | |
| MLB | 84.46±0.55 | 69.75±0.68 | 79.95±0.24 | 83.57±0.34 | 84.27±0.86 | 84.72±0.25 |
| MFB | 85.11±0.75 | 70.53±0.44 | 79.04±0.23 | 83.11±0.19 | 81.87±0.41 | 81.89±0.42 |
| TFN | 75.62±0.58 | 62.33±1.46 | 79.77±0.25 | 83.67±0.27 | 51.94±0.93 | 80.33±0.61 |
| LMF | 85.47±0.62 | 70.92±0.86 | 79.90±0.17 | 83.64±0.36 | 87.69±0.27 | **85.07±0.22** |
| PTP | 84.13±0.49 | 71.27±0.46 | 79.88±0.14 | 84.01±0.18 | 85.94±0.32 | 84.78±0.24 |
| **TEF (Ours)** | **90.26±1.41** | **74.41±0.60** | **82.12±0.17** | **86.47±0.10** | **89.81±0.20** | 84.11±0.84 |
| Basic fusion operators (Precision) | | | | | | |
| Add | 86.44±0.25 | 72.46±0.54 | 79.97±0.32 | 83.60±0.31 | 87.52±0.39 | 83.85±0.39 |
| Mul | 83.76±0.57 | 64.09±0.40 | 78.32±0.22 | 82.44±0.43 | 70.09±1.04 | 84.31±1.34 |
| Cat | 86.17±0.43 | 71.94±0.21 | 80.03±0.33 | 83.81±0.17 | 87.73±0.17 | 84.36±1.00 |
| Max | 86.72±0.53 | 71.13±0.56 | 80.25±0.27 | 84.19±0.36 | 81.22±0.51 | 83.13±1.11 |
| Avg | 86.80±0.68 | 72.54±0.59 | 79.83±0.36 | 83.78±0.28 | 87.34±0.28 | 84.58±0.96 |
| Advanced fusion operators (Precision) | | | | | | |
| MLB | 85.11±0.47 | 71.09±0.76 | 80.64±0.26 | 84.33±0.31 | 86.55±0.49 | 86.73±0.66 |
| MFB | 87.00±0.45 | 71.52±0.46 | 79.68±0.26 | 83.52±0.20 | 83.93±0.18 | 84.64±0.25 |
| TFN | 77.92±0.79 | 63.13±1.20 | 80.42±0.44 | 84.00±0.48 | 58.89±0.76 | 83.09±0.55 |
| LMF | 86.78±0.60 | 72.05±0.53 | 80.29±0.20 | 84.08±0.31 | 89.15±0.22 | 87.02±0.46 |
| PTP | 85.63±0.33 | 71.99±1.44 | 80.50±0.06 | 84.28±0.22 | 87.74±0.34 | 86.24±0.65 |
| **TEF (Ours)** | **92.33±1.23** | **75.17±0.55** | **82.55±0.32** | **86.65±0.18** | **93.01±0.19** | **91.02±1.02** |
| Basic fusion operators (F1) | | | | | | |
| Add | 85.56±0.14 | 72.24±0.59 | 79.71±0.25 | 83.45±0.28 | 85.20±0.49 | 82.92±0.24 |
| Mul | 82.91±0.47 | 63.52±0.62 | 77.13±0.39 | 81.93±0.68 | 67.92±1.07 | 83.53±0.12 |
| Cat | 85.12±0.09 | 71.82±0.29 | 79.90±0.26 | 83.68±0.18 | 85.44±0.20 | 83.57±0.50 |
| Max | 85.96±0.36 | 70.90±0.44 | 80.02±0.20 | 84.04±0.26 | 78.35±0.40 | 81.80±0.39 |
| Avg | 85.85±0.49 | 72.46±0.55 | 79.67±0.29 | 83.59±0.28 | 84.93±0.40 | 82.70±0.22 |
| Advanced fusion operators (F1) | | | | | | |
| MLB | 84.57±0.43 | 70.06±0.30 | 80.19±0.14 | 83.81±0.22 | 84.62±0.75 | 85.57±0.32 |
| MFB | 85.65±0.72 | 70.87±0.34 | 79.28±0.24 | 83.28±0.18 | 82.33±0.31 | 83.17±0.18 |
| TFN | 76.05±0.60 | 62.19±1.47 | 80.01±0.31 | 83.79±0.33 | 52.33±0.91 | 81.52±0.24 |
| LMF | 85.77±0.58 | 71.27±0.71 | 80.04±0.17 | 83.78±0.27 | 87.92±0.25 | 85.92±0.19 |
| PTP | 84.47±0.39 | 71.22±0.63 | 80.11±0.09 | 84.08±0.18 | 86.23±0.35 | 85.41±0.38 |
| **TEF (Ours)** | **90.96±1.30** | **74.65±0.56** | **82.29±0.24** | **86.54±0.10** | **90.87±0.12** | **87.26±0.38** |
| Basic fusion operators (Kappa) | | | | | | |
| Add | 88.28±0.08 | 69.40±0.78 | 75.55±0.29 | 80.08±0.33 | 87.52±0.41 | 80.63±0.23 |
| Mul | 86.20±0.42 | 60.05±0.72 | 72.27±0.46 | 78.17±0.85 | 72.28±0.90 | 81.50±0.24 |
| Cat | 87.94±0.18 | 68.88±0.54 | 75.81±0.33 | 80.32±0.21 | 87.96±0.20 | 81.38±0.56 |
| Max | 88.56±0.33 | 67.77±0.53 | 75.94±0.24 | 80.74±0.33 | 81.55±0.41 | 79.57±0.31 |
| Avg | 85.57±0.33 | 69.62±0.57 | 75.55±0.36 | 80.22±0.34 | 87.26±0.33 | 80.36±0.27 |
| Advanced fusion operators (Kappa) | | | | | | |
| MLB | 87.40±0.22 | 66.87±0.38 | 76.10±0.19 | 80.48±0.34 | 87.10±0.67 | 83.68±0.29 |
| MFB | 88.60±0.35 | 67.72±0.45 | 75.04±0.25 | 79.83±0.22 | 85.21±0.20 | 81.04±0.21 |
| TFN | 80.59±0.50 | 58.86±1.26 | 75.85±0.36 | 80.41±0.37 | 57.48±0.92 | 79.35±0.25 |
| LMF | 88.39±0.44 | 68.17±0.79 | 75.95±0.18 | 80.43±0.34 | 89.91±0.25 | 84.09±0.24 |
| PTP | 87.49±0.29 | 68.30±0.52 | 76.02±0.12 | 80.80±0.24 | 88.60±0.36 | 83.68±0.31 |
| **TEF (Ours)** | **93.10±1.29** | **71.97±0.64** | **78.63±0.28** | **83.74±0.12** | **92.40±0.12** | **84.39±0.49** |

