# OpenReview forum: "Trusted Multi-View Classification via Evolutionary Multi-View Fusion"
_ICLR.cc/2025/Conference — ICLR 2025 Poster_

### Official Review · Reviewer_a8gx · 2024-10-21

**Soundness:** 3
**Presentation:** 3
**Contribution:** 3
**Rating:** 6
**Confidence:** 4

**Summary:**

The paper addresses the issues present in pseudo-views generated by previous methods and proposes a solution. It introduces an evolutionary multi-view architecture search approach to generate high-quality fusion architectures to serve as pseudo-views, thus enabling adaptive selection of views and fusion operators.

**Strengths:**

The overall writing of the paper is smooth and easy to understand.

The experimental results are extensive and solid.

**Weaknesses:**

I'm not sure if the proposed pseudo-view generation method is innovative compared to previous neural architecture search (NAS) methods.

Additionally, I'm uncertain whether the proposed method can be applied to large-scale end-to-end multimodal datasets. This is because the method might be inefficient, especially for high-dimensional and complex multimodal data, and the number of operators in the NAS process may be insufficient to cover all necessary functions. Increasing the number of operators might also lead to a significant increase in computational cost. I am open to increasing the score upon receiving a well-justified explanation.

**Questions:**

Could you please describe in detail the differences from other multi-modal NAS methods, such as the DC-NAS (Divide-and-Conquer Neural Architecture Search for Multi-Modal Classification) mentioned in the manuscript?

At the methodological level, what is the significance of concatenating pseudo-views with the original views? In principle, it seems that the pseudo-view already contains the information of the original view.

---

> ### Author Response · Authors · 2024-11-20
>
> Thank you for professional comments. We have tried our best to address your questions and revised our paper by following suggestions from all reviewers.
>
> **W1 and Q1: Could you please describe in detail the differences from other multi-modal NAS methods, such as the DC-NAS (Divide-and-Conquer Neural Architecture Search for Multi-Modal Classification) mentioned in the manuscript?**
>
> Re: This is a very good comment. The main difference between TEF and other multi-modal NAS methods is that the pesudo view searched by TEF will be enhanced by concatenating the fusion architecture's decision output with each view within the fusion architecture while ones searched by other methods are not. The strategy alleviates the view imbalance problem, which allows us to obtain better quality pesudo view than other multi-modal NAS methods. This has been verified by the empirical results that TEF armed with the our obtained pesudo view achieves a better performance than others. The results are shown in Table 1 and 2, where TEF and TEF$_{1}$ denote that TEF is armed with our pseudo-view and one induced by other NAS methods, respectively. That is,  our pseudo view is enhanced by concatenating it with the fusion architecture's decision output, while the compared one is not. Table 1 is part of Table 5 in the original manuscript. Table 2 is provided by conducting more experiments.
>
> Table 1
> |Methods|AWA|NUA|Reuters5|Reuters3|VoxCeleb|YoutubeFace|
> |----|----|----|----|----|----|----|
> |TEF$_{1}$|91.60±0.20|73.71±0.48|79.93±0.54|85.33±0.41|91.44±0.14|76.68±1.44
> |TEF|93.26±1.25|75.12±0.57|82.26±0.23|86.49±0.10|92.41±0.14|86.02±0.41
>
> Table  2
> |Methods|PIE|HandWritten|Scene15|Caltech-101|CUB|Animal|NUS|
> |----|----|----|----|----|----|----|----|
> |TEF$_{1}$|95.81±0.78|98.75±0.24|75.74±0.65|95.11±0.51|94.17±0.56|89.16±0.28|45.18±0.23
> |TEF|97.57±0.78|99.65±0.13|78.01±0.48|96.04±0.32|95.92±0.62|90.18±0.08|47.80±0.30
>
> Additionally, we want to clarify that the paper aims to further release the potential of the trusted multi-modal learning methods (TMML) by introducing high-quality pesudo view into TMML and breaking its late-fusion limitation. We formulate the pesudo view generation as a population-based multi-view neural architecture problem in the framework of TMML. It is natural that other multi-modal NAS methods are easy to be coupled into TEF. We indeed do not pay more focus on the NAS design, however the pesudo view enhancement is novel and original.
>
> **W2:  Additionally, I'm uncertain whether the proposed method can be applied to large-scale … I am open to increasing the score upon receiving a well-justified explanation.**
>
> Re: As you mentioned, evaluating the fitness of all $N$ chromosome vectors indeed requires too much computing time at each generation. In fact, this still is open problem in population-based search methods. Fortunately, evolutionary multi-view learning has provided many accelerated solutions such as fitness caching (FC), search guided by core structure (CS). In this paper, we use these strategies to accelerate the TEF, and find  that the version of TEF armed with FC and CS is very high-efficiency. Obviously, new developments of the acceleration techniques in the evolutionary computation community can be readily integrated into the TEF, achieving a more higher-efficiency implementations of TEF if more effort is made.
> Table
> |Methods|FC|CS|Time|
> |----|----|----|----|
> |TEF$_{w/o}$|False|False|13.21h|
> |TEF|True|True| 2.46h|
>
> Additionally, it is hard to afford the overload that each modality backbone network is also searched. The researchers provide a tradeoff between effectiveness and efficiency by fixing the backbone network and only searching the fusion strategies, such as [1-5]. In this paper, we follow the manner for dealing with large-scale end-to-end multimodal datasets. Moreover, in the era of big modal, training time seem not be limitation for algorithm application. One algorithm can be thought to be good if the inference time is within the user's tolerance range in terms of efficiency.
> TEF is the kind of algorithm.
>
> Last, we would like to explain why chose evolutionary NAS. The NAS methods can be roughly three categories: gradient-based NAS (GNAS), reinforcement learning-based NAS (RLNAS), and evolutionary NAS (ENAS). GNAS requires a predefined search space and substantial memory, RLNAS relies on extensive computational resources, ENAS offers advantages such as global search capability, flexibility, and parallelization. This makes it particularly suitable for handling complex multi-view tasks and large search spaces.
>
> [1] Core-structures-guided multi-modal classification neural architecture search, IJCAI,2024
>
> [2] DC-NAS: Divide-and-conquer neural architecture search for multi-modal classification, AAAI,2024
>
> [3] BM-NAS: Bilevel multimodal neural architecture search, AAAI,2022
>
> [4] Deep multimodal neural architecture search, ACMMM,2020
>
> [5] MFAS: Multimodal fusion architecture search, CVPR,2019

---

> ### Author Response · Authors · 2024-11-20
>
> **Q2: At the methodological level, what is the significance of concatenating pseudo-views with the original views? In principle, it seems that the pseudo-view already contains the information of the original view**
>
> Re:  The pseudo-view already contains the information of the original view. However, its quality still limited due to the view imbalance problem.  Moreover, the searched pesudo view will be optimized with other multiple views for trusted fusion. In this process, the problem of imbalanced multi-view learning exacerbates because as the searched pesudo view contains a disproportionate amount of information compared to individual views. Hence, we enhance each view within the fusion architecture by concatenating the fusion architecture's decision output with its respective view.
>
> The strategy is inspired by cortico-thalamocortical circuit, where the sensory processing mechanisms of different modalities modulate one another via the non-lemniscal sensory thalamus.
> Besides, from the perspective of characteristic among early fusion and late fusion, multi-view features are typically integrated through neural networks to achieve late fusion representations. However, this approach may result in the loss of certain raw information within individual views. On the other hand, early fusion methods combine features at an earlier stage but face challenges such as feature heterogeneity and high sample complexity. To address the limitations of both approaches, we innovatively extract late fusion information from the fusion architecture discovered in the first stage. This information is then incorporated into the second stage, where it is concatenated with the original views to generate pseudo-views. This strategy bridges the gaps in both early and late fusion methods, leading to improved results.
>
>
> Based on above analysis, we concatenate pseudo-views with the original views. The experiment results (See Table3  that is part of Table 5 in the original manuscript) show that the strategy is very effective.  TEF$_{2}$ directly uses the pseudo-view without concatenating with the original views for trustworthy fusion.
>
> Table 3
> |Methods|AWA|NUA|Reuters5|Reuters3|VoxCeleb|YoutubeFace|
> |----|----|----|----|----|----|----|
> |TEF$_{2}$|89.81±0.42|73.93±0.52|80.11±0.51|84.79±0.46|62.74±0.20|74.47±2.18
> |TEF|93.26±1.25|75.12±0.57|82.26±0.23|86.49±0.10|92.41±0.14|86.02±0.41
>
> Thank you again for professional comments. Please kindly let us know if you have any follow-up questions or areas needing further clarification. Your insights are valuable to us, and we stand ready to provide any additional information that could be helpful.

---

> ### Author Response · Authors · 2024-11-28
>
> Dear Reviewer a8gx,
>
> We thank you for your thorough review of our paper and for providing constructive feedback that has significantly contributed to its improvement. Your insights have been invaluable in helping us refine our work.
>
> We sincerely hope that our responses have sufficiently addressed the issues you highlighted in your review and follow-up comments. As the author-reviewer discussion period approaches its end, please do not hesitate to let us know if there is anything further we could do to improve your impression and final rating of our work.

---

> ### Author Response · Authors · 2024-12-02
>
> Dear Reviewer a8gx,
>
> We are pleased to address your concerns. And thank you very much for raising the score.
>
> Best regards,
>
> the authors.

---

### Official Review · Reviewer_jfaY · 2024-11-01

**Soundness:** 2
**Presentation:** 3
**Contribution:** 2
**Rating:** 3
**Confidence:** 4

**Summary:**

The paper presents TEF (Trusted multi-view classification via Evolutionary Fusion), aiming to address challenges in multi-view classification such as limited view interaction and learning imbalance. TEF utilizes evolutionary neural architecture search to create pseudo-views and applies a balanced fusion strategy. The method shows potential improvements over existing approaches in certain comparisons.

**Strengths:**

1. Combines evolutionary NAS with multi-view classification, offering a novel approach.
2. The methodology is clearly explained with effective visuals.

**Weaknesses:**

1. The emphasis on learning imbalance within pseudo-view-guided multi-view classification may not be as critical as the authors suggest. As long as the overall contributions of different views are balanced, focusing on the specific imbalance of the pseudo-view might be unnecessary. More evidence or justification would be helpful to show that this imbalance significantly impacts performance.

2. The experimental comparisons may not ensure fairness, as it seems that the benchmarks used in this study do not align with those originally used in previous TMVC methods (e.g. ECML [1]). This raises concerns about consistency and fairness. Clarifying whether all models were evaluated under similar conditions would improve the reliability of the results.

3. The claimed robustness of TEF in handling noisy or uncertain real-world data is not well-demonstrated. More experiments simulating practical conditions are needed to validate this aspect.

[1] Reliable conflictive multi-view learning. AAAI 2024

**Questions:**

1. Can you provide more justification or empirical evidence showing that learning imbalance within pseudo-view-guided multi-view classification significantly impacts model performance?

2. The benchmarks used in the paper seem different from those used in prior TMVC studies. Were the TMVC methods evaluated under the same conditions as your proposed TEF? Clarifying this would address concerns about fairness in experimental comparisons.

3. Can you show additional experiments or real-world examples demonstrating TEF's effectiveness in handling noisy or uncertain data?

4. Actually, the method seems to rely on aligning pseudo-view generation with target domain performance using validation labels. However, when samples are noisy, there appears to be no specific strategy to mitigate this noise. How does your approach show advancements over existing methods, especially given that Evidential Deep Learning has known limitations in fusion frameworks that could risk performance degradation?

---

> ### Author Response · Authors · 2024-11-20
> **Reply Questions 1,2**
>
> Thank you for professional comments. We have tried our best to address your questions and revised our paper by following suggestions from all reviewers.
>
> **W1,Q1: Can you provide more justification or empirical evidence showing that learning imbalance within pseudo-view-guided multi-view classification significantly impacts model performance?**
>
> Re: One of the main topics in multi-view learning is how to effectively integrate heterogeneous information from different views. However, although multi-view learning aids in comprehensively understanding the world by integrating information from various senses, most models often fail to satisfactorily achieve multi-view collaboration and do not fully utilize all views. Additionally, while it is anticipated that multiple input views would enhance model performance, we actually find that although multi-view models outperform single-modality models, their potential is still not fully realized [1] [2] [3]. Particularly, the introduction of pseudo-views can exacerbate this imbalance issue, where easily trainable views may suppress the potential of pseudo-views, leading the entire model to suboptimal performance. In Figure 3 of the paper, we demonstrate the performance difference before and after addressing the imbalance issue with pseudo-views, showing noticeable changes. Furthermore, we have added results from seven datasets commonly used in TMVC, leading to a consistent conclusion that the learning imbalance in pseudo-view guided multi-view classification has a significant impact on model performance.
>
> | Dataset     | PIE        | HandWritten | Scene15    | Caltech101 | CUB      | Animal   | NUS$_1$    | YouTube   | Reuter5   | Reuter3  | AWA     | NUS$_2$    | VoxCeleb |
> |-------------|------------|-------------|------------|------------|----------|----------|----------|-----------|-----------|----------|---------|----------|----------|
> | Imbalance   | 95.81±0.77 | 98.75±0.23  | 75.74±0.65 | 95.11±0.51 | 94.16±0.56| 89.16±0.27|45.00±0.23| 76.68±1.44| 79.93±0.54| 85.33±0.41|91.33±0.41| 73.71±0.2| 91.44±0.14|          |
> | Balance     | **97.57±0.78** | **99.64±0.13**  | **78.00±0.48** | **96.04±0.32** | **95.92±0.62**| **90.18±0.08**| **47.52±0.30**| **86.02±0.41**| **82.49±0.23**| **86.49±0.10**| **93.28±1.25**| **75.12±0.57**| **92.41±0.12**|          |
>
>
> **W2,Q2: The benchmarks used in the paper seem different from those used in prior TMVC studies. Were the TMVC methods evaluated under the same conditions as your proposed TEF? Clarifying this would address concerns about fairness in experimental comparisons.**
>
> Re: We have clearly stated in Appendix A.4 that the other TMVC methods and TEF are evaluated under the same conditions. Specifically, these methods utilize the same data processing workflows, dataset partitioning strategies, and consistently employ a 128-dimensional view dimension. Additionally, we have meticulously adjusted the parameter settings of the other TMVC methods to ensure they achieve better performance. Based on these fair comparison conditions, the results ensure the fairness of the evaluation.
>
> To fully demonstrate the advantages of our method and further verify its fairness, we also conducted evaluations on seven benchmark datasets commonly used by other TMVC methods. Using the same data partitioning strategy as Xu et al. (2024), we repeated the experiments ten times randomly and calculated the average results and standard deviations. The results show that our method exhibits significant advantages across all seven datasets.
>
> | Method    | PIE           | HandWritten   | Scene15       | Caltech101    | CUB           | Animal        | NUS$_1$           |
> |-----------|---------------|---------------|---------------|---------------|---------------|---------------|---------------|
> | EDL [4]   | 86.25±0.89    | 96.90±0.16    | 52.76±0.54    | 73.35±1.73    | 86.22±0.36    | 84.30±1.76    | 22.33±0.64   |
> | DCCAE [5]    | 81.96±1.04    | 95.45±0.35    | 74.62±1.52    | 89.56±0.41    | 85.39±1.36    | 82.72±1.38  | 35.75±0.48 |
> | CPM-Nets [6]  | 88.53±1.23    | 94.55±1.36    | 67.29±1.01    | 90.35±2.12    | 89.32±0.38    | 87.40±1.12    | 35.37±1.05  |
> | DUA-Nets [7]  | 90.56±0.47    | 98.10±0.32    | 68.23±0.11    | 93.43±0.34    | 80.13±1.67    | 78.65±0.55    | 33.98±0.34  |
> | TMC [8]       | 91.85±0.23    | 98.51±0.13    | 67.71±0.30    | 92.80±0.50    | 90.57±2.96    | 79.31±0.43    | 35.18±1.55   |
> | TMDL-OA [9]   | 92.33±0.36    | 99.25±0.45    | 75.57±0.02    | 94.63±0.04    | 95.43±0.20    | 87.05±0.28    | 34.39±0.44   |
> | RCML [10]    | 94.71±0.02    | 99.40±0.00    | 76.19±0.12    | 95.36±0.38    | 94.50±1.13    | 84.01±6.3     | 34.04±0.27   |
> | RMVC [11]   | 91.18±0.24    | 98.51±0.04    | 73.05±0.24    | 88.73±0.60    | 93.18±0.47    | 87.67±0.17    | 34.68±0.32   |
> | Ours      | **97.57±0.78**    | **99.64±0.13**    | **78.00±0.48**    | **96.04±0.32**    | **95.92±0.62**    | **90.18±0.08**    | **47.52±0.30**    |

---

> > ### Comment · Reviewer_jfaY · 2024-11-24
> > **How about Q3、Q4？**
> >
> > I cannot find the reply to question 3,4

---

> > > ### Author Response · Authors · 2024-11-24
> > > **Reply Questions 3,4**
> > >
> > > Dear Reviewer jfaY,
> > >
> > > We have replied the question 1,2 and the question 3,4 in two windows,respectively. However, we find that only one  window is shown when it is read from the mobinephone. Two windows will be shown when it is read from the computer.
> > >
> > > We are sorry for this issue. For your convenience, we also show the question 3,4 as follows.
> > >
> > > **W3, Q3: Can you show additional experiments or real-world examples demonstrating TEF's effectiveness in handling noisy or uncertain data?**
> > >
> > > Re:  Thank you for your insightful query, which enables us to more comprehensively demonstrate the advantages of the TEF framework. Concerning the seven conflicting datasets, they have been structured in accordance with Xu et al., 2024 [10]. TEF was rigorously tested on each dataset through ten iterations to ensure statistical robustness, with both mean values and standard deviations reported. These results substantiate TEF's effectiveness in managing datasets with inherent noise or uncertainty. As demonstrated, TEF consistently delivers superior performance, even under challenging conditions involving noisy or uncertain data. Notably, on the NUS dataset, TEF achieved a performance that was 12.08% higher than the second-best method, and on the Scene15 dataset, it surpassed the next best by 15.3%.
> > >
> > >
> > > | Method    | PIE         | HandWritten | Scene15     | Caltech-101 | CUB         | Animal      |  NUS$_1$         |
> > > |-----------|-------------|-------------|-------------|-------------|-------------|-------------|-------------|
> > > | EDL [4]       | 21.76±0.67  | 57.25       | 14.28±0.24  | 55.74±0.12  | 53.75±0.42  | 30.71±0.27  | 18.07±0.28  |
> > > | DCCAE [5]     | 26.89±1.10  | 82.85±0.38  | 25.97±2.86  | 60.90±2.32  | 63.57±1.28  | 64.30±2.11  | 32.12±0.52  |
> > > | CPM-Nets [6]  | 53.19±1.17  | 83.34±1.07  | 29.63±1.12  | 66.54±2.89  | 68.82±0.17  | 64.83±0.35  | 29.20±0.81  |
> > > | DUA-Nets [7]  | 56.45±1.75  | 87.16±0.34  | 26.18±1.31  | 75.19±2.34  | 60.53±1.17  | 62.46±1.12  | 31.82±0.43  |
> > > | TMC [8]       | 61.65±1.03  | 92.76±0.15  | 42.27±1.61  | 90.16±2.40  | 73.37±2.16  | 64.85±1.19  | 33.76±2.16  |
> > > | TMDL-OA [9]   | 68.16±0.34  | 93.05±0.45  | 48.42±1.02  | 90.63±2.35  | 74.43±0.36  | 64.62±0.15  | 32.44±0.26  |
> > > | RCML [10]      | 84.00±0.14  | 94.40±0.05  | 56.97±0.52  | 92.36±1.48  | 76.50±1.15  | 62.67±0.81  | 31.19±0.22  |
> > > | RMVC [11]      | 76.47±3.43  | 94.75±0.75  | 49.83±2.23  | 80.56±0.71  | 72.78±0.42  | 66.00±0.59  | 24.62±3.19  |
> > > | Ours      |  **86.76±0.49**  | **98.70±0.31**  | **72.27±0.43**  | **93.42±0.58**  | **77.41±0.47**  | **70.61±0.12**  | **45.84±0.31**
> > >
> > >
> > > **Q4:  Actually, the method seems to rely on aligning pseudo-view generation with target domain performance using validation labels. However, when samples are noisy, there appears to be no specific strategy to mitigate this noise. How does your approach show advancements over existing methods, especially given that Evidential Deep Learning has known limitations in fusion frameworks that could risk performance degradation?**
> > >
> > > Re: It is important to clarify that all methods were configured using 80% of the data as the training set and 20% as the test set for training and testing purposes. Considering the potential data leakage issues associated with Neural Architecture Search (NAS), we further divided the training set into a training set and a validation set to prevent data leakage. After completing the search and fusion architecture, we merged the training and validation sets into a new training set to proceed to the credible fusion stage. This practice is consistent with all traditional Multi-view Convergence (TMVC) methods and does not rely on using validation labels to align pseudo-view generation with target domain performance. On datasets R3, R5, and seven additional noise-inclusive datasets, TEF showed significant improvements compared to other methods, thanks to the introduction of pseudo-views. Even when multi-view data contain noise, as they still represent different descriptions of the same object, we can enhance performance through early fusion and feature interaction, which is evident from the experimental results. Additionally, pseudo-views are generated through a search process that effectively excludes some highly noisy views. Regarding the known limitations of Evidential Deep Learning in fusion frameworks, which could potentially degrade performance, the core issue is that they employ late fusion methods that overlook early feature interactions. Introducing a pseudo-view that has undergone sufficient feature interaction can significantly address these shortcomings.
> > >
> > > Thank you again for professional comments. Please kindly let us know if you have any follow-up questions or areas needing further clarification. Your insights are valuable to us, and we stand ready to provide any additional information that could be helpful.

---

> ### Author Response · Authors · 2024-11-28
>
> Dear Reviewer  jfaY,
>
> We thank you for your thorough review of our paper and for providing constructive feedback that has significantly contributed to its improvement. Your insights have been invaluable in helping us refine our work.
>
> We sincerely hope that our responses have sufficiently addressed the issues you highlighted in your review and follow-up comments. As the author-reviewer discussion period approaches its end, please do not hesitate to let us know if there is anything further we could do to improve your impression and final rating of our work.
>
> Best regards,
>
> The authors.

---

> ### Comment · Reviewer_jfaY · 2024-12-02
>
> Thanks for your reply. I am sorry for the late response.
>
> Firstly, I found your response somewhat confusing due to the inadequate citation of sources. It was difficult for me to trace each reference to its corresponding article, which caused considerable difficulty in understanding your points.
>
> To better understand the development in this field, I carefully reviewed the literature related to [1]. You just mentioned early feature interactions, so I would like to ask how the method presented in [2] differs from yours. I was also surprised to find that you cited experimental data from this work but did not provide a direct comparison with it. Additionally, there are other relevant works, such as [3], which seem to bear some relation to your study. Given its publication timeline, I am unsure whether this work was available before your submission. For now, I believe it would be sufficient for you to clarify the similarities and differences between [2] and your approach.
>
> I was quite perplexed by your mention of the evolutionary optimization pseudo-view method. The adaptive function is directly determined by downstream performance (classification accuracy), which means it is entirely supervised. This raises a tricky question: could there be a more detailed division of the training set, or further "evolution" across multiple dimensions? In any case, since the evolution is ultimately supervised, it seems that this work is quite incremental. If the pseudo-view evolution were based on other self-supervised or unsupervised metrics, I believe it would offer more insightful contributions.
>
> [1] Xu et al. Reliable Conflictive Multi-view Learning, AAAI 2024
>
> [2] Huang H et al. Trusted Unified Feature-Neighborhood Dynamics for Multi-view Classification, arXiv'24
>
> [3] Fu et al. Core-Structures-Guided Multi-Modal Classification Neural Architecture Search, IJCAI'24

---

### Official Review · Reviewer_fWQm · 2024-11-02

**Soundness:** 3
**Presentation:** 4
**Contribution:** 4
**Rating:** 8
**Confidence:** 5

**Summary:**

This research is significant in the field of trustworthy fusion and addresses two main challenges in current methods. First, many approaches overlook feature-level interactions, resulting in suboptimal performance. Second, high-quality pseudo-views exacerbate multi-view imbalance. The authors propose the TEF method, which utilizes evolutionary multi-view architecture search to generate high-quality pseudo-views, enabling adaptive selection of viewpoints and fusion operations. Experimental results demonstrate that this strategy significantly enhances TEF's performance on complex multi-view datasets, particularly in cases with three or more viewpoints. Evaluation results confirm the superiority of the proposed method compared to existing techniques.

**Strengths:**

1. The authors first review previous work and identify its limitations.
2. They innovatively introduce neural architecture search to generate high-quality pseudo-view architectures, providing a detailed description that effectively addresses the issue of insufficient feature interaction. Although this technique is time-consuming, it offers an effective solution.
3. The use of concatenation operations to solve the imbalance between multi-views is actually quite interesting, appearing simple yet effective.
4. Extensive experiments also demonstrate the effectiveness of TEF.

**Weaknesses:**

See in the questions.

**Questions:**

1. The paper indicates that introducing a single pseudo-view is effective; have there been any attempts to use multiple pseudo-views to further enhance the results?
2. Is this method a general concept? Can other trustworthy fusion methods also adopt similar strategies to improve model performance?
3.What do you believe is the fundamental reason behind achieving state-of-the-art results?
4.Regarding the time-consuming issue caused by using the NAS algorithm, while the paper suggests some solutions, I would still like to know if the authors have plans for further research in the future to tackle more complex data or deeper network issues.

---

> ### Author Response · Authors · 2024-11-20
>
> Thank you for professional comments. We have tried our best to address your questions and revised our paper by following suggestions from all reviewers.
>
> **Q1: The paper indicates that introducing a single pseudo-view is effective; have there been any attempts to use multiple pseudo-views to further enhance the results?**
>
> Re: Yes, we have considered this and conducted experiments on the NUS dataset. The table below shows the differences between not introducing any pseudo-views and introducing 1, 2, 3, or 4 pseudo-views. Introducing a single pseudo-view typically results in better performance, while introducing two or more pseudo-views does not significantly improve outcomes. This is because introducing one optimal pseudo-view has already enabled sufficient feature interaction, and additional pseudo-views may lead to conflicts or feature redundancy.
>
> | Number of Pseudo-Views | Accuracy (%) |
> |------------------------|--------------|
> | 0   | 72.73        |
> | 1   | 75.12        |
> | 2   | 75.24        |
> | 3   | 75.19        |
> | 4   | 75.09        |
>
> **Q2:  Is this method a general concept? Can other trustworthy fusion methods also adopt similar strategies to improve model performance?**
>
> Re: This method can be viewed as a plug-and-play module, suitable for conventional reliable multi-view classification methods. By introducing a pseudo-view generation framework, other methods can effectively address the common issues of traditional multi-view methods: these methods mostly use late fusion techniques, tend to merge the local features of each view, and overlook the global feature interactions between views. The introduction of pseudo-views not only compensates for this deficiency but also significantly enhances the overall performance of the model by integrating global features. Therefore, other credible fusion methods could consider adopting a similar strategy to achieve comparable improvements in model performance. We integrated this method into the classic TMC framework, and the results displayed in the table below illustrate its effectiveness. Compared to the original TMC, the enhanced TMC_TEF model demonstrates significant performance improvements across seven datasets. Specifically, the improvements on the PIE, Scene15, and NUS_1 datasets are 4.18%, 12.7%, and 10.43%, respectively.
>
> | Method    | PIE           | HandWritten   | Scene15       | Caltech101    | CUB           | Animal        | NUS$_1$           |
> |-----------|---------------|---------------|---------------|---------------|---------------|---------------|---------------|
> | TMC [1]       | 91.85±0.23    | 98.51±0.13    | 67.71±0.30    | 92.80±0.50    | 90.57±2.96    | 79.31±0.43    | 35.18±1.55   |
> | TMC_TEF   | **96.03±1.35**    | **98.86±0.13**     | **80.41±0.51**     | **94.71±0.57**     | **95.25±0.79**     | **89.36±0.19**    | **46.51±0.27**   |
>
> [1] Trusted Multi-View Classification 2021 ICLR
>
> **Q3:  What do you believe is the fundamental reason behind achieving state-of-the-art results?**
>
> Re:  The main reasons include: First, most credible fusion methods employ a late fusion strategy, which limits the interaction of information between views, consequently leading to suboptimal utilization of multi-view data. To overcome this challenge, we adopted an evolutionary multi-view architecture search method, creating a high-quality fusion architecture to serve as a pseudo-view. This architecture abandons the traditional constraints of late fusion, allowing features to fully interact and effectively aggregate global features rather than just focusing on local features. Second, the introduction of pseudo-views may exacerbate the imbalance issue in multi-view learning, as the pseudo-views contain more information relative to individual views, which could potentially weaken model performance. To mitigate this issue, we enhanced each view within the fusion architecture by concatenating the decision outputs of the fusion architecture with their corresponding views, thereby enhancing the efficacy of the pseudo-views and further optimizing the overall performance of the model.

---

> ### Comment · Reviewer_fWQm · 2024-11-23
>
> The authors have addressed my concerns by conducting more experiments and providing more explanations. I would like to increase my score.

---

> > ### Author Response · Authors · 2024-11-24
> >
> > Thank you for raising the score. We are glad to hear that our revisions and feedback have successfully addressed your initial concerns.

---

### Official Review · Reviewer_por9 · 2024-11-04

**Soundness:** 4
**Presentation:** 3
**Contribution:** 4
**Rating:** 8
**Confidence:** 5

**Summary:**

This paper illustrates two main issues in existing trustworthy fusion methods: the lack of feature interaction in late fusion and the insufficient attention given to multi-view imbalance, which leads to inadequate training and suboptimal performance. To address this, the authors innovatively propose an evolutionary computation adaptive method that introduces high-quality pseudo-views, significantly enhancing the performance of trustworthy fusion methods on complex multi-view datasets. Furthermore, the authors present an effective solution for view imbalance and conduct extensive experiments to validate its effectiveness.

**Strengths:**

1. The paper presents the motivation for the research in detail through illustrations, and the structure of the paper is logical and well-written.
2. The paper demonstrates considerable innovation, as the authors first identify two key challenges in trustworthy multi-view fusion and provide effective solutions.
3. The authors validate the effectiveness of their method through extensive experiments, with results showing significant advantages over existing trustworthy and non-trustworthy fusion methods across five evaluation metrics on six datasets.

**Weaknesses:**

1. I am interested in your concatenation operation. Why is the late-stage fusion information only concatenated with the pseudo-views and not with the original views? What effects would concatenation have?
2. Why did you choose evolutionary computation for generating pseudo-views? Have you considered gradient neural architecture search or reinforcement learning?
3. Can early fusion features not assist in training? Why was the final choice made to use late-stage features to address view imbalance?

**Questions:**

See in the weakness.

---

> ### Author Response · Authors · 2024-11-20
> **Reply to the Reviewer por9**
>
> Thank you for professional comments. We have tried our best to address your questions and revised our paper by following suggestions from all reviewers.
>
> **W1: I am interested in your concatenation operation. Why is the late-stage fusion information only concatenated with the pseudo-views and not with the original views? What effects would concatenation have?**
>
> Re:
> In the first stage, evolutionary neural architecture search generates an optimal feature interaction fusion architecture. In the second stage, the method produces a high-quality pseudo-view. However, the training difficulties between the single-view training architecture and the feature interaction fusion architecture exacerbate the imbalance problem, leading to insufficient training of the feature fusion architecture. Auxiliary post-processing features generated in the first stage can assist in the training of the second stage to achieve the desired performance. It is evident from the table that the use of feature concatenation and its placement significantly impacts the performance of the TEF architecture. TEF^0 introduces a pseudo-view framework, TEF^1 introduces pseudo-views without concatenation, TEF^2 performs concatenation before the classification layer, and TEF^3 concatenates within the original view. Comparing TEF^3 and TEF^1, TEF^3 shows a significant performance improvement on all datasets, including a 9.44% gain on YouTubeFace. Furthermore, the performance of TEF^3 surpasses that of TEF^2, better preserving and utilizing multi-view information when concatenating at the original view stage rather than after view fusion but before classification.
> | Methods | AWA      | NUS      | Reuter5  | Reuter3  | VoxCeleb | YoutubeFace |
> |---------|----------|----------|----------|----------|----------|-------------|
> | $TEF^0$|88.59±0.25|72.73±0.30|79.60±0.56|84.23±0.35|73.13±0.15|71.18±2.27|
> | $TEF^1$|91.60±0.20|73.71±0.48|79.93±0.54|85.33±0.41|91.44±0.14|76.68±1.44|
> | $TEF^2$|93.16±1.21|75.06±0.66|81.14±0.63|86.45±0.18|92.06±0.14|85.35±0.62|
> | $TEF^3$|**93.26±1.25**|**75.12±0.57**|**82.26±0.23**|**86.49±0.10**|**92.41±0.12**|**86.02±0.41**|
>
>
> **W2:  Why did you choose evolutionary computation for generating pseudo-views? Have you considered gradient neural architecture search or reinforcement learning?**
>
> Re: We have comprehensively considered three neural architecture search (NAS) methods and ultimately chose evolutionary neural architecture search (eNAS) for the following reasons: Compared to gradient-based NAS, which requires predefined search spaces and substantial memory, and reinforcement learning-based NAS, which relies on extensive computational resources, evolutionary NAS offers advantages such as global search capability, flexibility, and parallelization (Liang et al., 2021). This makes it particularly suitable for handling complex multi-view tasks and large search spaces.
>
> **W3:  Can early fusion features not assist in training? Why was the final choice made to use late-stage features to address view imbalance?**
>
> Re: We conducted an analysis on the impact of extracting features from various layers for view enhancement, using the Reuters3 dataset. Specifically, we concatenated features from the last four layers of the fusion architecture to assess their final effect in the TEF (Tensor Ensemble Framework). Assuming the fusion architecture consists of n layers, we extracted features from the n-th, (n-1)-th, (n-2)-th, and (n-3)-th layers, and also set a baseline that did not use any features for comparison. The experimental results, as shown in Figure 6, indicate that the closer the layer is to the final one, the better the performance. For instance, concatenating features from the n-th layer resulted in a 0.59% improvement in accuracy compared to concatenating features from the (n-3)-th layer. This is because the deepest layers of the fusion architecture contain the most complex and advanced representations of the model, which can capture more in-depth patterns and semantic information from the data. Enhancing the original view with multi-view representations post-fusion helps improve the expressiveness of the view and prevents information loss during the fusion process. This approach facilitates better outcomes in multi-view fusion architectures and alleviates the multi-view imbalance issue to some extent.
>
> | Metric   | None  | n-3   | n-2   | n-1   |
> |----------|-------|-------|-------|-------|
> | Accuracy | 84.02 | 84.47 | 84.68 | **85.14** |
> | Recall   | 83.55 | 83.85 | 84.67 | **85.27** |
> | Precision| 84.73 | 84.83 | 85.15 | **85.69** |
> | F1       | 84.14 | 84.34 | 84.91 | **85.48** |
> | Kappa    | 83.44 | 83.59 | 83.85 | **84.41** |
>
> Thank you again for professional comments. Please kindly let us know if you have any follow-up questions or areas needing further clarification. Your insights are valuable to us, and we stand ready to provide any additional information that could be helpful.

---

> > ### Comment · Reviewer_por9 · 2024-11-26
> >
> > I would like to thank the authors for the detailed answers, which have addressed my concerns. After carefully reviewing the other reviews, I will maintain my current ratings.

---

> > > ### Author Response · Authors · 2024-11-28
> > >
> > > Dear Reviewer por9,
> > >
> > > Thank you for the approval of our work contribution, and give it 8: accept, good paper.
> > >
> > > We are glad to hear that our revisions and feedback have successfully addressed your initial concerns.

---

### Author Response · Authors · 2024-11-27

Dear Reviewers,

Thank you again for your valuable feedback. We have carefully addressed your comments in the rebuttal and revised the manuscript accordingly.

We kindly invite you to review the updates, and please let us know if you have any further questions or suggestions. Your time and insights are greatly appreciated.

Best regards,

Authors

---

### Meta-Review · Area_Chair_kvpt · 2024-12-15

**Metareview:**

This paper introduces the Enhancing Trusted Multi-View Classification via Evolutionary Multi-View Fusion (TEF) approach. During the rebuttal phase, the authors effectively addressed key concerns related to experimental fairness, robustness in noisy scenarios, computational efficiency, and generalizability. As a result, three out of four reviewers provided strong ratings of 8, 6, and 6, demonstrating the overall positive reception of the work.

The first-stage pseudo-view generation approach is regarded as sound, as it ensures no leakage of testing samples, thereby maintaining the integrity and independence of the evaluation process. The core contribution of this paper lies in enhancing trusted multi-view classification by advancing feature-level interaction between views—a less-explored but critical aspect in this domain.

Thus, acceptance is recommended.

**Additional Comments On Reviewer Discussion:**

This paper improves feature-level interactions in trusted multi-view classification and merits acceptance.

---

### Decision · Program_Chairs · 2025-01-22

Accept (Poster)